# Zeb1-Hdac2-eNOS circuitry identifies early cardiovascular precursors in naive mouse embryonic stem cells

Chiara Cencioni[1,2], Francesco Spallotta[1], Matteo Savoia[1,3], Carsten Kuenne[4], Stefan Guenther[4], Agnese Re[2], Susanne Wingert[5], Maike Rehage[5], Duran Sürün[5], Mauro Siragusa[6], Jacob G. Smith[7], Frank Schnütgen[5], Harald von Melchner[5], Michael A. Rieger[5], Fabio Martelli[8], Antonella Riccio[7], Ingrid Fleming [6], Thomas Braun[9], Andreas M. Zeiher[10], Antonella Farsetti[2,10] & Carlo Gaetano [1,11]

Nitric oxide (NO) synthesis is a late event during differentiation of mouse embryonic stem cells (mESC) and occurs after release from serum and leukemia inhibitory factor (LIF). Here we show that after release from pluripotency, a subpopulation of mESC, kept in the naive state by 2i/LIF, expresses endothelial nitric oxide synthase (eNOS) and endogenously synthesizes NO. This eNOS/NO-positive subpopulation (ESNO+) expresses mesendodermal markers and is more efficient in the generation of cardiovascular precursors than eNOS/NO-negative cells. Mechanistically, production of endogenous NO triggers rapid Hdac2 S-nitrosylation, which reduces association of Hdac2 with the transcription repression factor Zeb1, allowing mesendodermal gene expression. In conclusion, our results suggest that the interaction between Zeb1, Hdac2, and eNOS is required for early mesendodermal differentiation of naive mESC.

[1] Division of Cardiovascular Epigenetics, Department of Cardiology, Goethe University, Theodor-Stern-Kai 7, 60590 Frankfurt am Main, Germany. [2] National Research Council, Institute of Cell Biology and Neurobiology (IBCN), Via del Fosso di Fiorano 64, 00143 Rome, Italy. [3] Institute of Medical Pathology, Università Cattolica di Roma, Largo Francesco Vito 1, 00168 Rome, Italy. [4] ECCPS Bioinformatics and deep sequencing platform, Max Planck Institute for Heart and Lung Research, Ludwigstrasse 43, 61231 Bad Nauheim, Germany. [5] LOEWE Center for Cell and Gene Therapy and Department of Medicine, Hematology/Oncology, Goethe University, Theodor-Stern-Kai 7, 60590 Frankfurt am Main, Germany. [6] Institute for Vascular Signalling, Centre for Molecular Medicine, Goethe University, Theodor-Stern-Kai 7, 60590 Frankfurt am Main, Germany. [7] MRC Laboratory for Molecular Cell Biology, University College London, Gower St, Kings Cross, London WC1E 6BT, UK. [8] Molecular Cardiology Laboratory, IRCCS-Policlinico San Donato, Via Morandi 30 San Donato Milanese 20097 Milan, Italy. [9] Department of Cardiac Development and Remodeling, Max-Planck-Institute for Heart and Lung Research, Ludwigstrasse 43, 61231 Bad Nauheim, Germany. [10] Internal Medicine Clinic III, Department of Cardiology, Goethe University, Theodor-Stern-Kai 7, 60590 Frankfurt am Main, Germany. [11] Laboratorio di Epigenetica, Istituti Clinici Scientifici Maugeri, Via Maugeri 4, 27100 Pavia, Italy. These authors contributed equally: Chiara Cencioni, Francesco Spallotta. Correspondence and requests for materials should be addressed to C.C. (email: chcencioni@gmail.com) or to A.F. (email: antonella.farsetti@cnr.it) or to C.G. (email: carlo.gaetano@icsmaugeri.it)

The gaseous product of nitric oxide synthases (NOS), nitric oxide (NO), has been reported among the compounds able to control *Nanog* expression and function[1,2], thus suggesting a possible role in the regulation of mESC fate[2]. A body of literature, in fact, established NO as an essential factor for cardiovascular precursor generation during mESC cardiovascular lineage commitment[3–5]. This effect seems to depend on experimental conditions and, at least in one report, it has been described that low doses of NO may repress differentiation[6]. Of note, all these observations were made by adding exogenous sources of NO to the mESC medium or expressing a wild-type eNOS in cells cultured in the presence of leukemia inhibitory factor (LIF)[2,4,5,7]. Most commonly, the endogenous NO production has been described as a relatively late event occurring in mESC after release from pluripotency and associated with the expression of functional NOS isoforms[8]. Nevertheless, previous experiments established that, in vivo, NO might also be an active molecule during the very early phases of preimplantation embryo attachment[9]. Surprisingly, in this condition the NO concentration has never been established and, how the endogenous production of NO might be regulated at very early embryonal differentiation stages is currently unknown. The "ground state-like" (GS) culture system may help to provide more information about this aspect.

NO is crucial for some biological functions including the regulation of epigenetic enzymes[7,10–13]. Specifically, NO controls the activity of histone deacetylases (Hdacs) at multiple levels[7,10–13]. For example, it inhibits the nuclear function of class I Hdac2 by direct S-nitrosylation[10,13] and facilitates the nuclear translocation of class II Hdacs, including Hdacs 4 and 5, depending on protein phosphatase 2A (PP2A) activation in response to cyclic guanosine monophosphate (cGMP) production[7,11]. Hdacs are master regulators of the differentiation process and in particular, Hdac2, coupled to its cognate corepressor Hdac1, is involved in maintenance of mESC pluripotency through cooperation with Oct4 and Nanog[14]. Recently, the role of these two Hdacs has been further dissected[15,16]. Although Hdac1 has been found important for early cardiovascular commitment of mESC, while Hdac2 could be apparently dispensable[15], later work[16] demonstrated the relevance of both class I Hdacs in mESC proliferation and pluripotency. Interestingly, Hdac1 and Hdac2 are differentially regulated by numerous post-translational modifications (PTMs)[17]. Hdac1 has been reported phosphorylated, acetylated, SUMOylated, and ubiquitinylated, while Hdac2, similarly to Hdac6[18], was also found S-nitrosylated. Surprisingly, in skeletal muscle, Hdac2 S-nitrosylation seemed to have negative effects on the function of both Hdacs[10]. Moreover, the diverse Hdac-specific PTMs have been found relevant to the pathophysiology of cardiovascular diseases[19]. Whether Hdac2 S-nitrosylation might play a role during the early mesendodermal commitment of naive mESC and whether, in the presence of NO, Hdac2 might affect Hdac1 function, remains unexplored.

The Zinc finger E-box binding homeobox 1 and 2 transcription repressor factors (Zeb1; Zeb2) are essential players in the epithelial-to-mesenchymal transition (EMT), a process able to reorganize epithelial cells to become migratory mesenchymal cells[20–22]. Zeb factors, in fact, directly target E-cadherin transcription, determining its downregulation, a hallmark of EMT[21,23]. Interestingly, genetic inactivation studies indicated a crucial role of Zeb1 also during embryonic development[24–27]. Specifically, Zeb1 knock-out mice exhibited alterations compatible with neuroectodermal defects, neural crest EMT impairment, and aberrant bone formation[26,27]. In recent studies, the sporadic insurgence of vascular changes and hemorrhages were also reported[25]. Hence, Zeb1 genetic inactivation determined complex phenotypes affecting the development of ectodermal and mesendodermal structures[24–27].

According to prior work, the miR-200 family is induced by NO donors and contributes to Zeb2 downregulation during mESC mesendodermal differentiation[28]. Although evidence implicates Zeb2 in the repression of mesendodermal commitment in mESC[28], this phenomenon has not been mechanistically addressed nor has the role of Zeb1 been established in this context. Recently, Zeb1 has been reported associated with class I Hdacs as part of a larger repressor complex[23,29]. This evidence suggests the presence of a molecular circuitry involving mir-200, Zeb factors, Hdacs and pluripotency genes contributing to keep mESC undifferentiated[30]. This hypothesis is supported by the role of Zeb1 in the maintenance of cancer stem cells[21], an effect mediated by transcriptional repression of miRs involved in EMT progression control[22]. Nevertheless, from all these studies little information emerges about Zeb1 and its role as a negative determinant of embryonic differentiation.

In the present manuscript, we describe Zeb1 orchestrating, together with Hdac2, a transcriptional repression network of important mesendodermal-associated genes whose expression is facilitated in a Zeb1-reduced environment. Among the repressed genes we found eNOS, a determining factor for early mesendodermal commitment of naive mESC[8]. In this context, cells expressing eNOS and synthesizing NO, in the presence of S-nitrosylated Hdac2 and low levels of Zeb1, have been identified as early precursors that, once isolated, were able to recapitulate the mesendodermal/cardiovascular lineage differentiation program.

## Results

**NO activates mesendodermal commitment in naive mESC.** Unless otherwise indicated, all experiments in mESC were performed in GS condition[31] according to the 2i/L protocol. As expected, in comparison to culture medium with serum and LIF alone (defined as standard medium, SM), GS cultures appeared as relatively spherical colonies with homogenous morphology and better-defined borders (Supplementary Fig. 1a). In this condition, mESC expressed higher levels of pluripotency-associated proteins including Klf4, Nanog, Oct4, and Sox2 (Supplementary Fig. 1b). To compare the biological responses to NO of mESC grown in SM vs. GS, cells were released for 24 h from inhibitors and cultured in complete medium supplemented with 10% fetal bovine serum in the absence (differentiation medium, DM) or presence of the NO donor diethylenetriamine NONOate[32] (DM plus DETA/NO, NO). This treatment changed the shape of cell clusters characterized by enlarged flat colonies with irregular margins (Supplementary Fig. 1a, DM vs. NO). To assess NO responsiveness at the molecular level, we evaluated the expression of the miR-200 family members. In SM, miR-200b, miR-200a, and miR-429 were induced after LIF withdrawal (DM), an effect enhanced in NO (Supplementary Fig. 1c, middle upper graph)[28]. Of note, miR-200c and miR-141, belonging to the miR-200 family cluster 2 and mapping to mouse chromosome 6 (Supplementary Fig. 1c, left lower cartoon), were not significantly upregulated by NO compared to controls (Supplementary Fig. 1c, middle lower graph). Conversely, in cells released from GS in the presence of NO, miR-200 family member induction was robust and significant for all cluster components (Supplementary Fig. 1c, right graphs). The relative primiR expression was also analyzed in mESC released from GS with or without NO. Specifically, the primiR transcribed from cluster 1 was significantly induced by NO while that of cluster 2 was less sensitive being already significantly expressed in cells cultured in DM alone (Supplementary Fig. 1d). The reason why cluster 2 was less responsive to NO signaling is currently unknown. However, published data suggest that the two clusters might be differentially regulated at

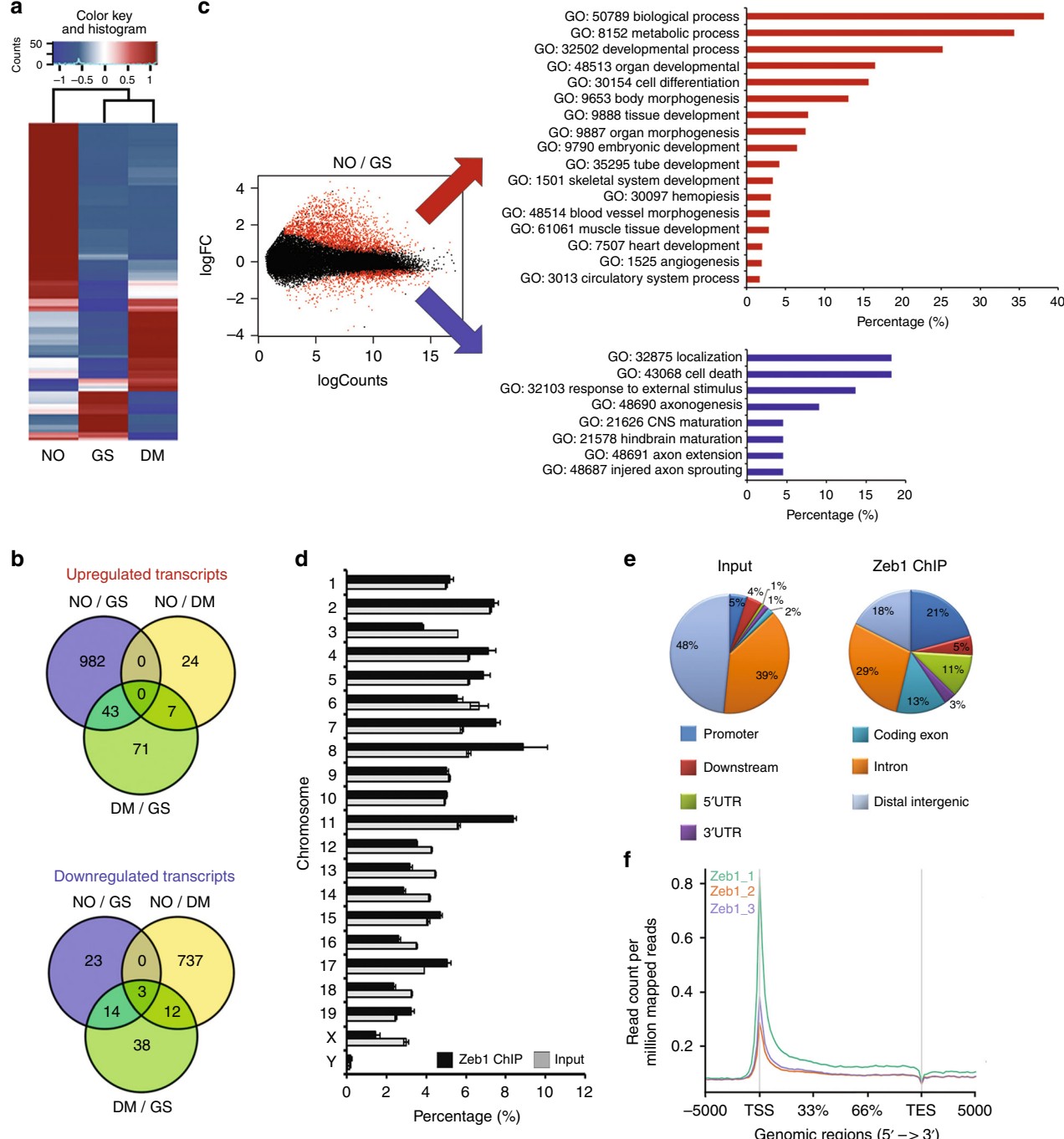

**Fig. 1** Identification of NO-responsive transcripts and Zeb1-associated genomic regions. **a** Heatmap showing the 50 most differentially regulated genes in mESC cultured for 24 h in GS, DM, or NO identified by total RNA sequencing analysis (n = 3 each group). Red and blue represent over- and under-expressed genes, respectively. **b** Venn diagrams depicting the distribution of upregulated (upper panel) and downregulated (lower panel) unique or common transcripts among GS, DM, or NO conditions. **c** Left panel: MA plot of differentially regulated genes expressed in mESC cultured for 24 h in NO compared to GS condition. Red dots show genes with a p adjusted value <0.05. Right panels: Gene ontology analysis of NO-differentially regulated transcripts between mESC cultured in GS and NO. Upregulated genes depicted in the upper panel, red bar graph; downregulated genes in the lower panel, blue bar graph. **d** Distribution of chromatin–associated Zeb1 regions over mouse chromosomes. Black bars: percentage of all chromatin/Zeb1-associated regions; gray bars: background derived from input samples (n = 3). **e** The pie chart illustrates the distribution of chromatin-associated Zeb1 regions (Zeb1-ChIP) over specific genomic features depicted as follows: (i) promoters encompassing 3000 bp upstream of the transcription start site (TSS) (blue sector); (ii) downstream regions encompassing 3000 bp from transcription end site (TES) (red sector); 5′ UTRs (green sector) and 3′ UTRs (purple sector) as well as coding exons (turquoise sector) and introns (orange sectors) relative to the gene bodies; iii) distal intergenic regions (light blue sectors). Input and Zeb1-ChIP represented in the left and right pie chart respectively. **f** Gene body profile depicting Zeb1-ChIP regions distribution in association with TSS or TES produced by ngsplot

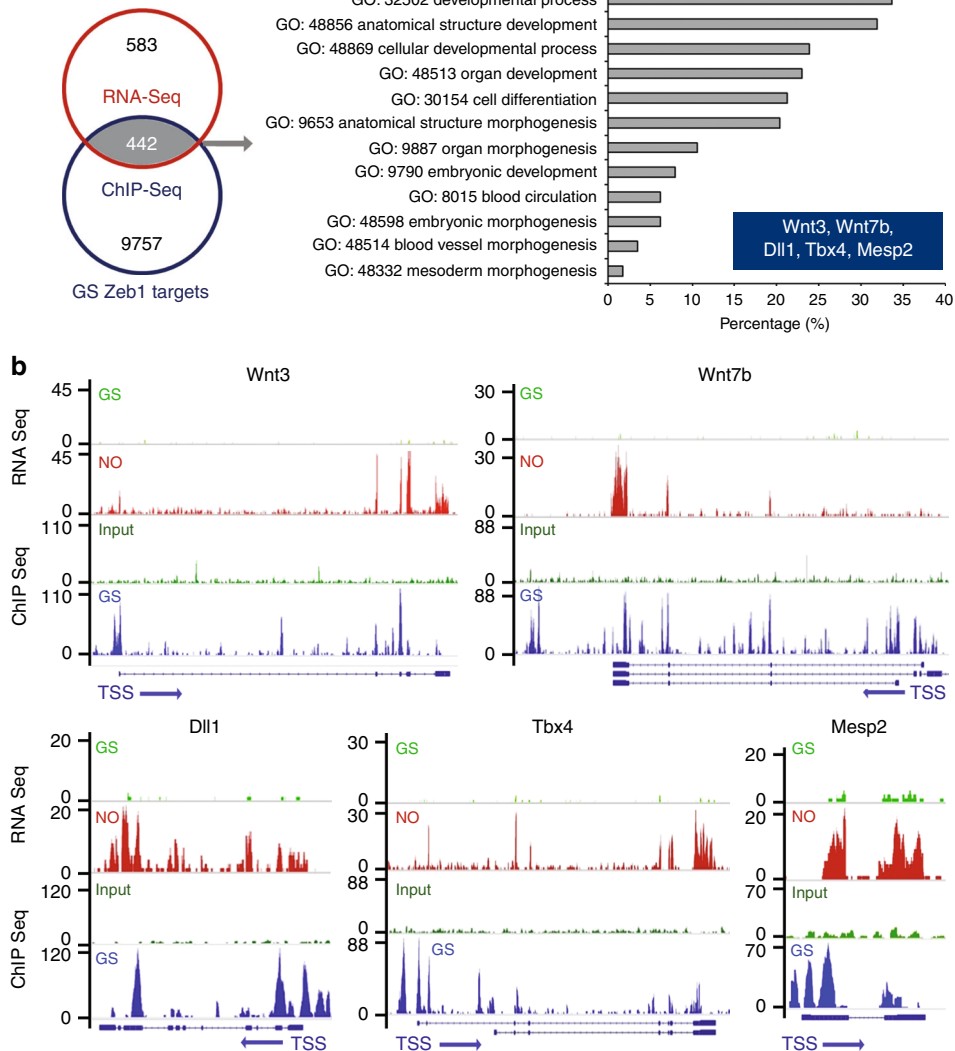

**Fig. 2** Zeb1 binds chromatin at NO-responsive mesendodermal gene promoters. **a** Left panel: Venn diagram showing intersection between 1025 NO-upregulated genes (NO up-genes) identified by RNA Seq analysis (red circle) and 10199 gene promoters in which Zeb1 has been found associated with chromatin at TSS in cells cultured in GS condition (GS Zeb targets; blue circle). 442 common targets were found (gray area). Right panel: The gray arrow points to the result of gene ontology analysis based on the 442 common targets at the intersection between RNA-seq and ChIP-seq (left panel). The five most represented genes, alias Mesp2, Wnt3, Wnt7b, Tbx4, and Dll1, listed in the blue colored inset. **b** Sashimi plots showing detailed peak positioning as determined by RNA-seq and ChIP-seq data for each selected mesendodermal gene. Specifically, for each plot, RNA-seq data related to GS (light green) or NO (red) culture conditions and ChIP-seq data related to GS for input (dark green) and Zeb1 localization (blue) were represented

transcription level by mechanisms involving DNA methylation and Polycomb complex function[33]. Furthermore, p53 positively regulates the miR-200 family transcription[34,35] and NO is among the signaling molecules regulating p53[35]. Specifically, DETA/NO stabilizes p53 level and increases its acetylation at lysine residue 379 (Supplementary Fig. 1e). In this condition, chromatin immunoprecipitation (ChIP) showed p53 bound to the miR-200 cluster 1 promoter region (Supplementary Fig. 1f).

The effect of DETA/NO on naive mESC has been further assessed comparing GS, DM, and NO by transcriptome analysis. Specifically, 39,178 differentially expressed genes (DEGs) were found after analysis of GS and DM or NO for 24 h. After pairwise comparison of GS/DM, NO/GS, and NO/DM, 883, 2286, and 1815 genes, respectively, were found differentially expressed at more than ±1 $\log_2$ fold change (base-mean >5, FDR <0.05). Among those, the 50 most significant DEGs of each pair (GS/NO, GS/DM, NO/DM), based on the multiple testing adjusted $p$-value

criteria, were selected resulting in 123 different genes (Supplementary Table 2). DESeq normalized counts of the selected sequences were averaged per condition and depicted as a heatmap using a hierarchical clustering generated by Pearson correlation of the $z$-score (Fig. 1a). The result revealed that NO treatment induced important changes in mESC transcriptome. Further, the overlapping extent of DEGs among the experimental conditions was utilized to identify condition-specific regulated genes, and a Venn diagram was created to group RNAs into up- and downregulated transcripts (Fig. 1b), We identified 982 transcripts upregulated in NO compared to GS and 23 downregulated. The comparison of NO/DM alone identified 24 upregulated transcripts and 737 downregulated (Fig. 1b). To assign a role to NO-modulated transcripts, gene-ontology (GO) analysis was performed on significantly up- or downregulated RNAs. The interconnections among the upregulated transcripts indicated that these genes belonged to biological processes associated with

mesendodermal lineage. Specifically, blood vessel morphogenesis (GO-ID:48514), muscle tissue development (GO-ID:61061), and heart development (GO-ID:7507). In the same condition, the most downregulated genes were enriched in RNAs associated with neuroectodermal development, such as axonogenesis (GO-ID:50770), central nervous system maturation (GO-ID:21626), and hindbrain maturation (GO-ID:21578) (Fig. 1c). Altogether, these results implicate DETA/NO as a molecular inducer of mesendodermal differentiation. Intriguingly, bioinformatics analysis of NO/DM comparison further reinforced the indication that NO promotes mesendodermal commitment also by

repression of neurogenesis and epithelial differentiation (Supplementary Fig. 2).

**Zeb1 enrichment on mesendodermal gene chromatin.** Previous studies indicate Zeb1 and Zeb2 as targets of miR-200s[20,33] and Zeb2 negatively regulated by NO[28]. Remarkably, although Zeb1 has been implicated in self-renewal maintenance of cancer cells[22] little information is available about its role in GS or in NO-enriched medium. Hence, specific experiments were performed to assess the NO effect on Zeb1 in our system. Zeb1 protein levels were significantly reduced by DETA/NO in mESC released either

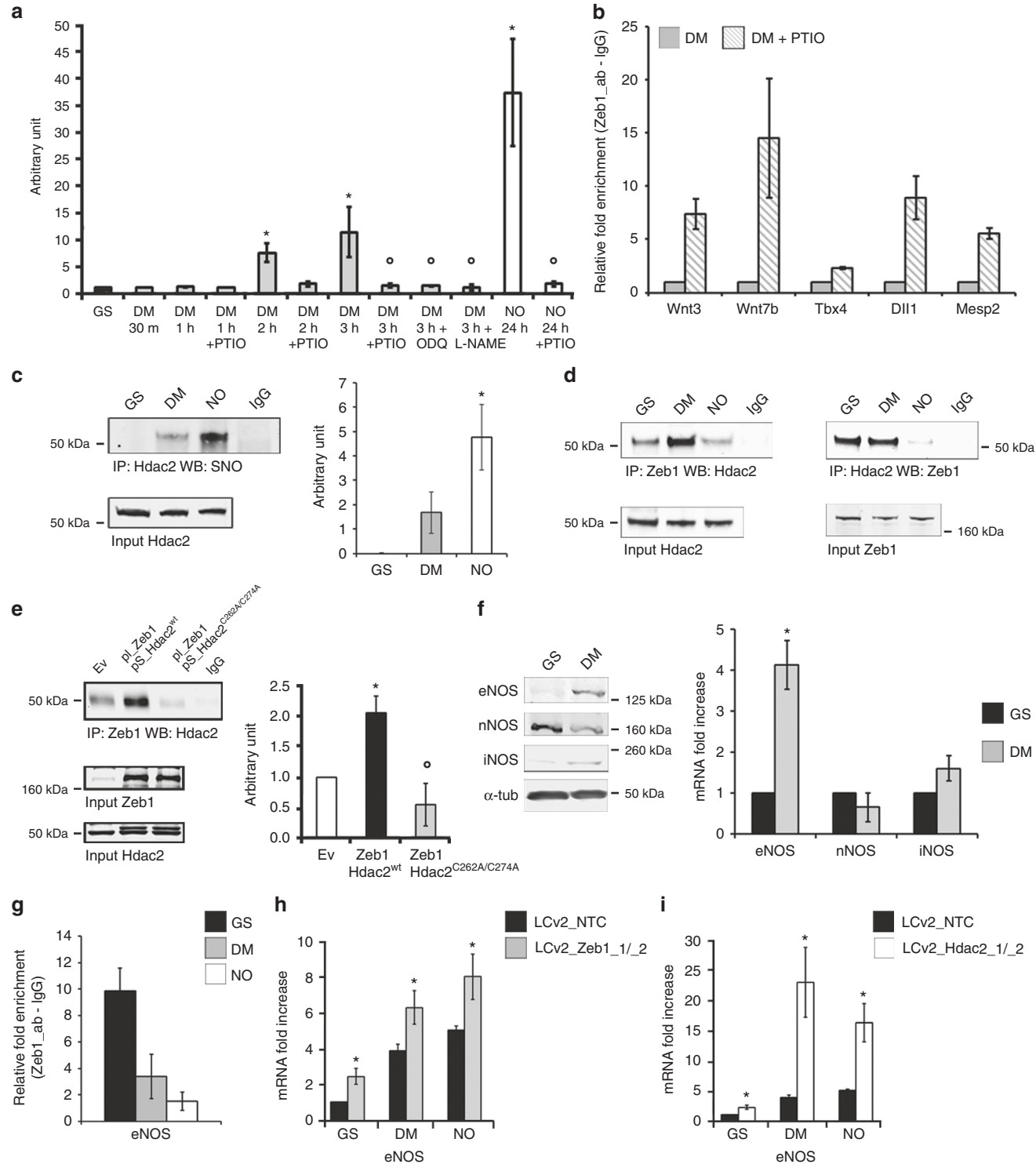

from SM or GS (Supplementary Fig. 3a). Confocal microscopy showed that Zeb1 is principally nuclear in mESC cultured in GS or DM (Supplementary Fig. 3b). Instead, in the presence of NO, Zeb1 partially relocated out of the nucleus (Supplementary Fig. 3b and insets therein). This evidence was confirmed by subcellular fractionation experiments showing enrichment of Zeb1 in the cytoplasm of DETA/NO-treated mESC (Supplementary Fig. 3c). To establish whether the NO-mediated Zeb1 down-modulation was regulated by induction of miR-200 family, we analyzed Zeb1 protein levels in mESC transfected with scramble or anti-miR-200b oligonucleotides. Western blot analysis showed that the inhibition of miR-200b, the most NO-sensitive miR-200 family member, rescued Zeb1 levels both in DM and in NO conditions (Supplementary Fig. 3d). Moreover, ChIP experiments revealed a feedback loop regulation between Zeb1 and miR-200 family in mESC (Supplementary Fig. 3e). Specifically, we observed an enrichment of Zeb1 binding on miR-200 family cluster 1 promoter in mESC kept in GS, suggesting active repression of the miR-200 family in self-renewal condition.

To investigate whether Zeb1 is involved in the regulation of mESC response to NO and in mesendodermal gene expression control, ChIP-sequencing (ChIP-seq) was performed in GS. A total of 17435 Zeb1-associated peaks, present in at least two out of three replicates, was considered. ChIP-seq reads were enriched on chromosomes (Chr.) 4, 5, 7, 8, 11, 15, 17 and 19 compared to the Input (Fig. 1d). Considering the genomic features, Zeb1 peaks occurred particularly at 5′ UTR, along promoters and in coding regions (Fig. 1e). However, the majority of these peaks, about 11225, clustered around the transcription start site (TSS ± 5 kb) of 10199 genes (Fig. 1f). Of note, about 54% of Zeb1 peaks contained the canonical human Zeb1 motif [CT]N[CT]ACCTG as registered in the JASPAR database [http://jaspar.genereg.net/], suggesting an important role for Zeb1 in the transcription gene regulation in naive mESC. To identify putative Zeb1 targets regulated by NO, an integrated RNA-seq/ChIP-seq dataset was generated (Fig. 2a). Specifically, we combined the 1025 NO-dependent genes identified by RNA-Seq analysis with the 10199 genes enriched for Zeb1 TSS binding as indicated by ChIP-seq. About 442 transcribed regions were identified (Fig. 2a, left) in which genes with a role in mesendodermal differentiation were overrepresented according to the GO analysis (Fig. 2a, right; Fig. 1b, c). This analysis identified five transcripts, Mesp2[36,37], Wnt3[38], Wnt7b[38], Tbx4[39], and Dll1[40], among those most represented and associated with a mesendodermal differentiation program (Fig. 2a, right graph, and inset therein). As quantitatively visualized, by the Sashimi plot analysis (Fig. 2b) at chromatin level, each selected gene showed a relative enrichment

in Zeb1 peaks in GS (blue peaks) compared to their input (dark green peaks, GS), as well as a significant increase of RNA peaks in NO (red peaks) compared to GS alone (light green peaks). These results suggest that: (i) NO activates a transcriptional program along the mesendodermal differentiation pathway; (ii) Zeb1 binding is enriched in the regulatory regions of mesendodermal-associated genes possibly contributing to their repression. The expression of the most represented NO-Zeb1-dependent genes was validated by qRT-PCR. DM condition determined a significant increase in the specific mRNA level of the mesendodermal-associated genes (gray bars). Notably, DETA/NO enhanced their expression (white bars) (Supplementary Fig. 4a). The presence of the NO scavenger 2-Phenyl-4,4,5,5-tetramethylimidazoline-1-oxyl 3-oxide (PTIO)[41] abolished the mesendodermal marker induction observed in DM condition suggesting an endogenous production of NO after release from self-renewal (Supplementary Fig. 4b). Gene-specific ChIP experiments were performed in the same condition. The Zeb1 enrichment was validated on the selected gene promoters in mESC kept in GS (black bars) (Supplementary Fig. 4c). Instead, in DM condition, DNA-bound Zeb1 became undetectable (gray bars), an effect independent from the presence of the NO donor (white bars).

Hdac1 and Hdac2 are known to associate with Zeb1 forming a transcription repression complex[23]. We wanted to investigate whether these members of the Hdac class I family were also associated with the promoter regions of the selected Zeb1 target genes. In this context, Hdac1 and Hdac2 showed a similar pattern of enrichment and both detached upon transferring mESC in DM or NO medium (Supplementary Fig. 4d, e). These results suggest that Zeb1 and Hdac1/2, other components of the Zeb1 repressor complex, are co-regulated during early differentiation of mESC. Further experiments were performed with LNA oligos designed against miR-200b. The results showed a significant reduction of mesodermal markers expression in the presence of NO suggesting that Zeb1 targeting is important for the effect of NO (Supplementary Fig. 4f).

**Endogenous NO synthesis occurs early during differentiation.**
After release from pluripotency, endogenous NO production occurred spontaneously during a time course from 1 to 24 h after changing medium (Supplementary Fig. 5a, left panel). However, in NO the number of DAF-positive cells was significantly higher. Interestingly, the normalized DAF fluorescence intensity was similar in both conditions (Supplementary Fig. 5a, right panel) suggesting that the amount of DETA/NO used determined a

**Fig. 3** Endogenous NO synthesis and Zeb1-Hdac2 complex formation. **a** cGMP quantification in GS (black bar), in DM ± PTIO, ODQ, L-NAME (gray bars) for 1, 2, or 3 h and in NO ± PTIO (white bars) for 24 h. (n = 3; *p < 0.05 DM vs. GS; °p < 0.05 treatments vs. DM). **b** Zeb1-chromatin binding analysis on: Wnt3, Wnt7b, Tbx4, Dll1, and Mesp2. Chromatins extracted in DM (gray bars) or DM + PTIO (striped bars). Data compared to IgG value (striped bars) after Input normalization (n = 3). **c** Left panel: representative immunoprecipitation (IP)/western blotting (WB) analysis of Hdac2 S-nitrosylation in GS (24 h), DM or NO (2 h). Right panel: densitometry (n = 3; *p < 0.05 NO vs. DM). Full-length blot provided in Supplementary Fig. 9a. **d** Reciprocal co-IP analysis showing Zeb1:Hdac2 (left) and Hdac2:Zeb1 (right) complex formation in GS, DM or NO (24 h; n = 3 each group). Full-length blot provided in Supplementary Fig. 9b and c. **e** Left panel: representative co-IP analysis showing Zeb1:Hdac2wt and Zeb1:Hdac2C262A/C274A complex formation compared to empty vector (Ev)-transfected HeLa cells. HeLa transfected with Ev or co-transfected with pCMV6_Zeb1/pCIG2_Hdac2wt or pCMV6_Zeb1/pCIG2_Hdac2C262A/C274A. Right panel: densitometry (n = 3; *p < 0.05 Zeb1:Hdac2wt vs. Ev; °p < 0.05 Zeb1:Hdac2C262A/C274A vs. Zeb1:Hdac2wt). Full-length blot provided in Supplementary Fig. 9d. **f** Left panel: representative WB analysis of eNOS, nNOS, and iNOS in GS or DM (n = 3). Loading control: α-tubulin. Full-length blot provided in Supplementary Fig. 9e. Right panel: eNOS, nNOS, and iNOS mRNA analysis in GS (black bars) or DM (gray bars). Data expressed as fold increase of GS after subtraction of the housekeeping gene p0 signal (n = 3; *p < 0.05 DM vs. GS). **g** Zeb1-chromatin binding analysis on eNOS promoter in GS (black), DM (gray), and NO (white). Data represented as fold increase of GS after input normalization (n = 3). **h, i** eNOS mRNA expression analysis in GS, DM, and NO prior (control vector LCv2_NTC; black bar) and after CRISPR/Cas9 inactivation of Zeb1 (**h**; LCv2_Zeb1_1/_2; gray bars) or CRISPR/Cas9 inactivation of Hdac2 (**i**; LCv2_Hdac2_1/_2; white bar). Data represented as fold increase compared to control after subtraction of the housekeeping gene p0 signal (n = 3; *p < 0.05 LCv2_Zeb1_1/_2 or LCv2_Hdac2_1/_2 vs. LCv2_NTC). Data represented as mean ± s.e.m. and analyzed by Kolmogorov–Smirnov test

release of NO comparable to that occurring spontaneously in untreated cells. To investigate whether the endogenous NO was sufficient to elicit biologically relevant effects, we determined cGMP production during a time course from 1 to 3 h from release. The cGMP production was assessed in GS or DM and in the presence or absence of PTIO, the guanylate cyclase inhibitor ODQ[42], the pan-NO inhibitor L-NAME[43], or DETA/NO (Fig. 3a). The results indicate a positive cGMP production detectable as early as 2 h after release from GS. This effect is the consequence of rapid endogenous NO synthesis as suggested by the ability of PTIO to abrogate cGMP production at all time-points tested and in the presence of DETA/NO. Similarly, ODQ and L-NAME both prevented cGMP synthesis (Fig. 3a). Consistently, the results of gene-specific ChIP experiments showed that Zeb1 detachment was reduced or abrogated by PTIO, associating the endogenous NO production (Fig. 3a) to regulation of the Zeb1/chromatin interaction (Fig. 3b) and activation of a lineage-specific gene expression. Interestingly, in NO Hdac2 was nitrosylated (Fig. 3c) and its association with Zeb1 was significantly decreased (Fig. 3d) suggesting that in this condition the Zeb1:Hdac2 complex formation was destabilized. To explore this aspect, a series of co-immunoprecipitations were performed by using an Hdac2 mutant[13] in which the nitrosyla-table Cysteines at positions 262 and 274 were substituted by Alanine (Hdac2$^{C262A/C274A}$). The result shows that this mutant, which is insensitive to nitrosylation, is unable to bind Zeb1, thus reproducing the effect of nitrosylation, suggesting that integrity of the residues at position 262 and 274 is important for Zeb1:Hdac2 association (Fig. 3e).

To provide evidence about the origin of the endogenous NO synthesis, a series of western blots were performed. We observed that nNOS was present in GS at both mRNA and protein level whereas eNOS and iNOS were not significantly expressed (Fig. 3f). In DM, however, a significant amount of eNOS protein became detectable while nNOS decreased rapidly. In this condition, iNOS was not expressed (Fig. 3f, left). In line with these observations, ChIP-Seq analysis performed in GS revealed a number of Zeb1 peaks distributed along the eNOS genomic locus whereas they were virtually absent from the iNOS gene and significantly less represented at the nNOS locus (Supplementary Fig. 5b). The Zeb1 enrichment in the eNOS genomic region strongly decreased in DM or NO (Fig. 3g). To investigate further, whether Zeb1 or Hdac2 were functionally involved in the regulation of eNOS, we generated two independent clones of mESC in which the endogenous genes were knocked-out by CRISPR-Cas9 technology (Supplementary Fig. 5c). In cells in which Zeb1 was knocked-out, expression of eNOS increased in all tested conditions (Fig. 3h). Similar results were obtained in cells in which Hdac2 was inactivated (Fig. 3i) suggesting negative regulation of eNOS gene expression by both Zeb1 and Hdac2. Notably, in those clones where Zeb1 or Hdac2 were knocked-out, expression of mesendodermal-associated genes became already detectable in GS (Fig. 4a, b).

**eNOS activity is important for mesendodermal gene expression.** In vivo studies indicated that production of NO is important for early stage embryo implantation in uterus[9,44]. However, NO synthesis or responsiveness, well documented in SM, occurs relatively late during the differentiation process[8]. Specifically, mESC cultured in SM are unable to synthesize NO during the very early phases of differentiation in association with

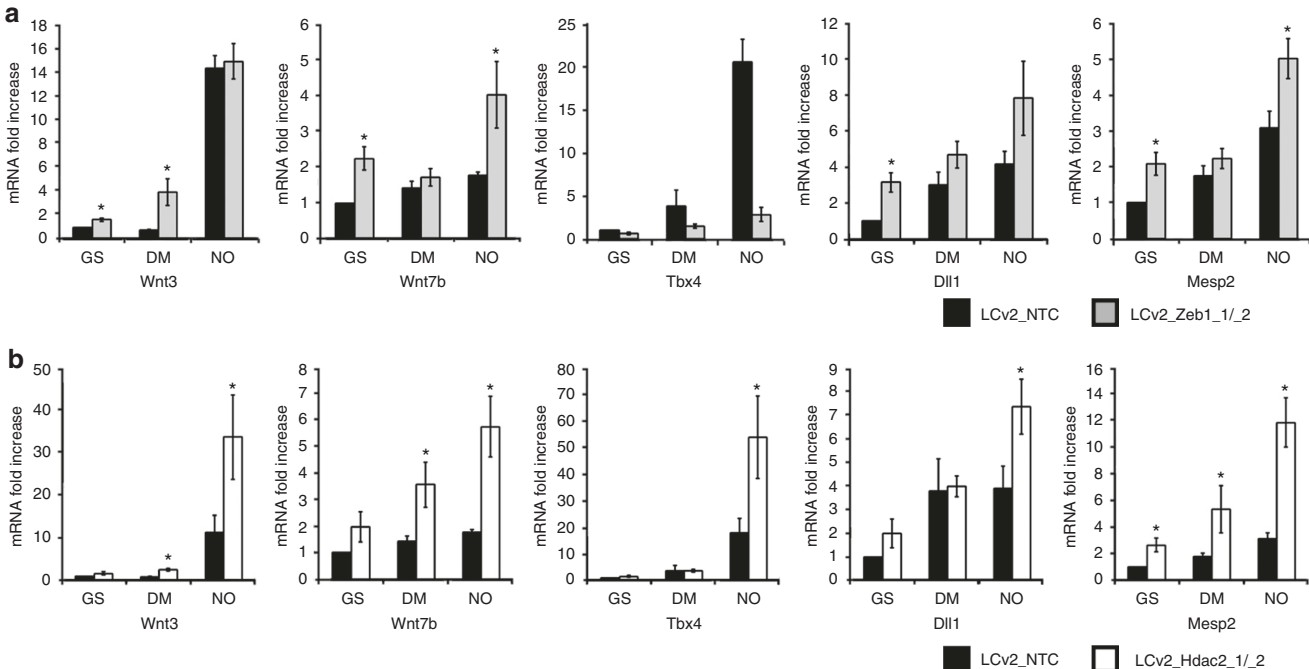

**Fig. 4** Evaluation of mesendodermal transcripts in mESC engineered by CRISPR-Cas9. **a** qRT-PCR analysis of mRNA expression relative to Wnt3, Wnt7b, Tbx4, Dll1, and Mesp2 transcripts in mESC cultured for 24 h in GS, DM, and NO prior (control vector LCv2_NTC; black bar) and after CRISPR/Cas9 inactivation of Zeb1 (LCv2_Zeb1_1/_2; gray bars). Data are shown as the mean of three independent experiments for each CRISPR/Cas9 vector ± s.e.m. represented as fold increase compared to control mESC after subtraction of the housekeeping gene p0 signal (*p < 0.05 LCv2_Zeb1_1/_2 vs. LCv2_NTC). **b** qRT-PCR analysis of mRNA expression relative to Wnt3, Wnt7b, Tbx4, Dll1, and Mesp2 transcripts in mESC cultured for 24 h in GS, DM, and NO prior (control vector LCv2_NTC; black bar) and after CRISPR/Cas9 inactivation of Hdac2 (LCv2_Hdac2_1/_2; white bars). Data are shown as the mean of three independent experiments for each CRISPR/Cas9 vector ± s.e.m. represented as fold increase compared to control mESC after subtraction of the housekeeping gene p0 signal (*p < 0.05 LCv2_Hdac2_1/_2 vs. LCv2_NTC). Data analyzed by Kolmogorov–Smirnov test

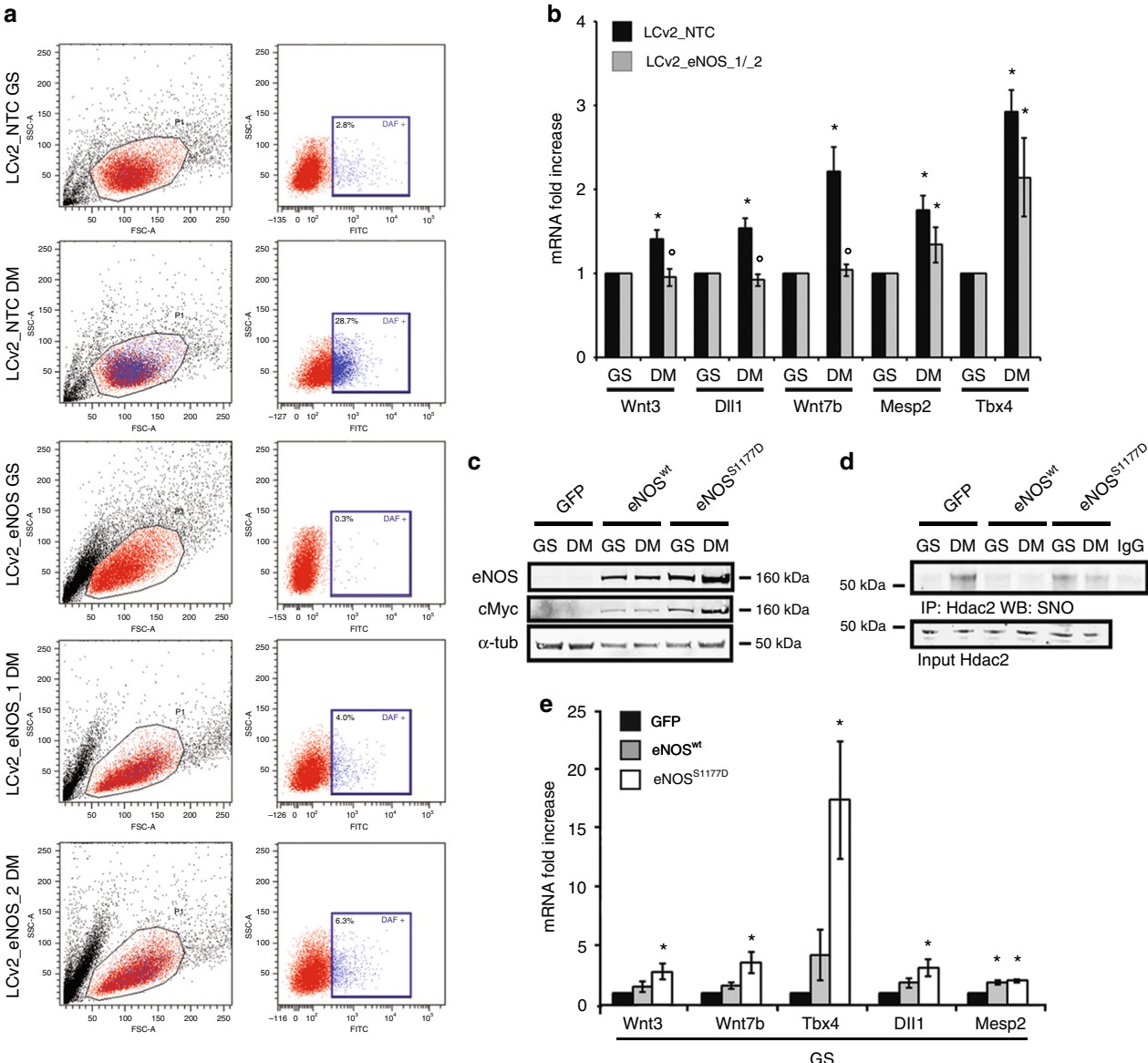

**Fig. 5** Role of eNOS during mesendodermal commitment. **a** Fluorescence activated cell sorter analysis of mESC cultured in GS and 2 h in DM prior (control vector LCv2_NTC) and after CRISPR/Cas9 inactivation of eNOS (LCv2_eNOS_1 or LCv2_eNOS_2). Left cytograms: representative scatter plots showing the forward (FSC-A) and side scatter (SSC-A) distribution of the mESC population. Right panels: representative dot plot showing the fluorescein isothiocyanate (FITC) and SSC-A distribution of the mESC population in the presence of DAF fluorescent probe. **b** qRT-PCR analysis of mRNA expression relative to Wnt3, Dll1, Wnt7b, Mesp2, and Tbx4 transcripts in mESC cultured for 24 h in GS and DM prior (control vector LCv2_NTC; black bar) and after CRISPR/Cas9 inactivation of eNOS (LCv2_eNOS_1/_2; gray bars). Data are shown as the mean of three independent experiments for each CRISPR/Cas9 vector ± s.e.m. represented as fold increase compared to control mESC after subtraction of the housekeeping gene p0 signal (*$p < 0.05$ DM vs. GS; °$p < 0.05$ LCv2_eNOS_1/_2 vs. LCv2_NTC). **c** Representative western blot performed with cell extracts obtained from mESC infected by Ad_GFP, Ad_eNOS[wt], or Ad_eNOS[S1177D] cultured for 24 h in GS or DM ($n = 3$ each group). The expression of eNOS and cMyc was evaluated. The signal from α-tubulin antibody was used as loading control. Full-length blot provided in Supplementary Fig. 10a. **d** Representative immunoprecipitation (IP) western blotting (WB) analysis of Hdac2 S-nitrosylation performed in mESC infected by Ad_GFP, Ad_eNOS[wt], or Ad_eNOS[S1177D] cultured in GS or after 2 h from release into DM ($n = 3$ each group). Full-length blot provided in Supplementary Fig. 10b. **e** qRT-PCR analysis of mRNA expression relative to Wnt3, Wnt7b, Tbx4, Dll1 and Mesp2 transcripts in mESC cultured for 24 h in GS prior (control Adeno virus Ad_GFP; black bar) and after infection by Ad_eNOS[wt](gray bars) or Ad_eNOS[S1177D] expressing the constitutive active eNOS (white bars). Data shown as the mean of three independent experiments ± s.e.m. represented as fold increase compared to Ad-GFP infected mESC after subtraction of the housekeeping gene p0 signal (*$p < 0.05$ vs. Ad_GFP). Data analyzed by Kolmogorov–Smirnov test

a relatively low level of *eNOS* expression and function[8]. On the contrary, cGMP synthesis occurring early in naive mESC released from GS, suggests that a mESC subpopulation could be responsible for the rapid NO synthesis. Therefore, the eNOS role in the early mesendodermal commitment of mESC has been explored in greater detail. In our hands, L-NAME but not ODQ prevented or

reduced the expression of eNOS and other mesendodermal-associated markers in mESC released from GS (Supplementary Fig. 5d). This observation indicates that an endogenous active synthesis of NO could be important for the observed gene regulation, while that of cGMP might not be relevant in the early commitment of mESC to mesendoderm. To functionally address

the role of eNOS in this context we generated two independent mESC clones inactivated by CRISPR/Cas9 technology for *eNOS* gene without affecting *nNOS* expression (Supplementary Fig. 5e). These clones were unable to synthesize significant amount of NO, compared to their controls, thus excluding an active involvement of nNOS in this process (Fig. 5a). Interestingly, in these clones, the expression of mesendodermal markers was significantly reduced (Fig. 5b). Conversely, the forced expression of a constitutively active form of *eNOS*, the mutant S1177D, (Fig. 5c) determined Hdac2 nitrosylation (Fig. 5d) and the spontaneous expression of genes associated with mesendodermal differentiation (Fig. 5e) becoming detectable in mESC kept undifferentiated. On the contrary, forced expression of Zeb1 (Supplementary Fig. 6a) reduced or prevented expression of mesendodermal markers in mESC trasferred into DM or NO (Supplementary Fig. 6b).

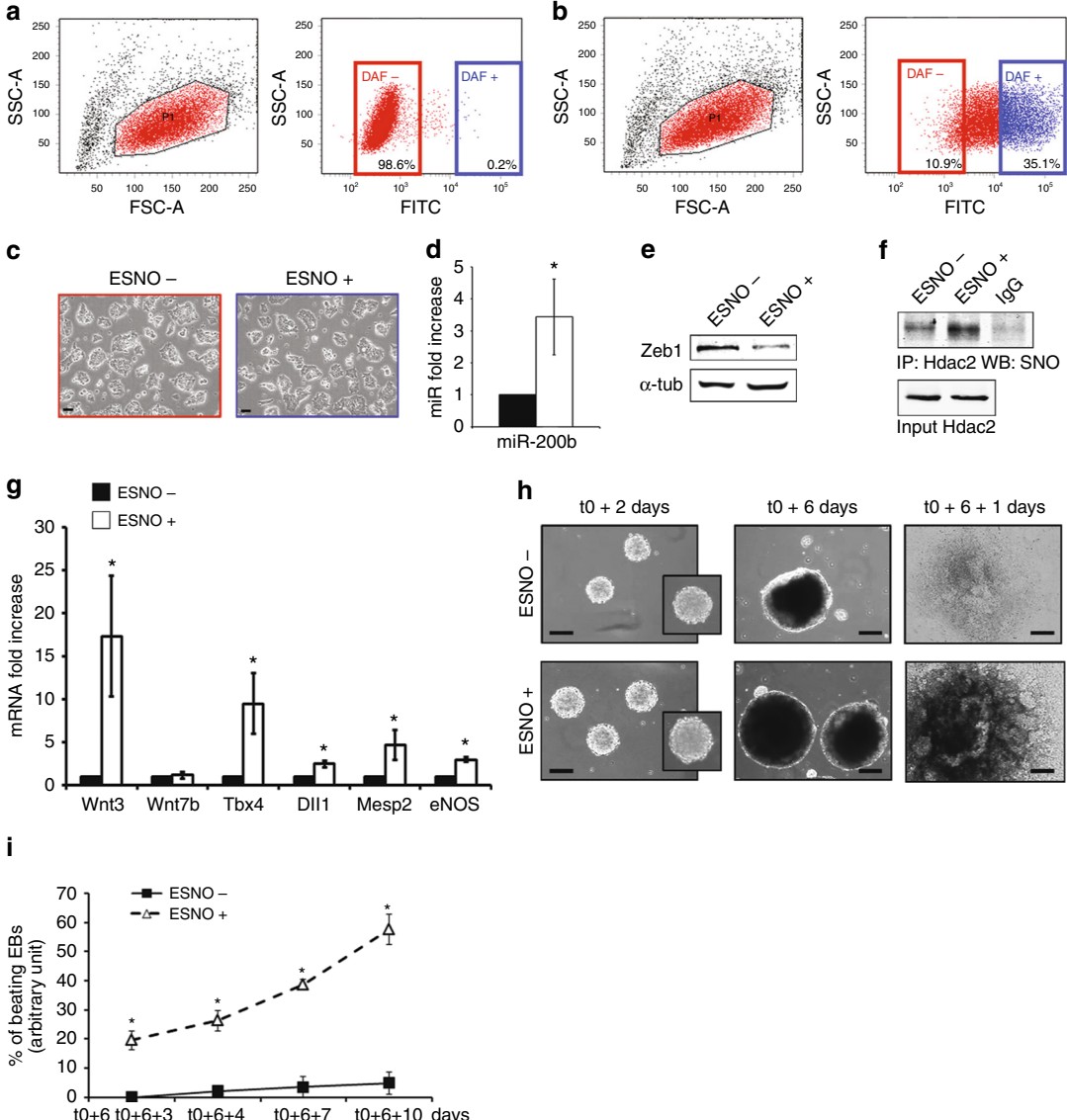

**Fig. 6** ESNO +/− subpopulation isolation and characterization. **a**, **b** FACS analysis. **a** Left: representative scatter plot showing forward (FSC-A) and side scatter (SSC-A) distribution in GS. Right: representative dot plot showing the fluorescein isothiocyanate (FITC) and SSC-A distribution in GS in the presence of DAF fluorescent probe. **b** Same as **a** related to 2 h DM. Each experiment: gating to separate ESNO+ (DAF+; blue dots) from ESNO− (DAF−; red dots) established manually ($n = 3$). **c** Representative phase contrast microscopy images of ESNO− (red upper panel) and ESNO+ (blue lower panel) cultured in DM. Scale bar 10 μm. **d** miR-200b expression analysis in ESNO+ (white bar) compared to ESNO− (black bar) at 24 h from sorting. Data shown as mean ± s.e.m. represented as fold increase compared to ESNO− after subtraction of the housekeeping gene p0 signal (*$p < 0.05$; $n = 3$). **e** Representative WB of Zeb1 in ESNO+ and ESNO− at 24 h from sorting ($n = 3$). Full-length blot provided in Supplementary Fig. 11a. **f** Representative IP/WB analysis of Hdac2 S-nitrosylation in ESNO+ and ESNO− after sorting ($n = 3$). Full-length blot provided in Supplementary Fig. 11b. **g** Wnt3, Wnt7b, Tbx4, Dll1, Mesp2, and eNOS expression analysis in ESNO+ (white bars) compared to ESNO− (black bars) at 24 h from sorting. Data shown as mean ± s.e.m. represented as fold increase compared to ESNO− after subtraction of the housekeeping gene p0 signal (*$p < 0.05$; $n = 3$). **h** Phase contrast microscopy images of ESNO− and ESNO+ derived EBs (upper and lower panels, respectively) taken at different stages of the EB formation process after sorting (t0): 2 and 6 days of hanging drop culture after t0 (t0 + 2 and t0 + 6, respectively) and the first day after plastic adherence following hanging drop culture (t0 + 6 + 1). Scale bar 30 μm. **i** Quantification of beating EBs generated by ESNO+ (open triangles) and ESNO− (closed squares) upon a time course from three (t0 + 6 + 3) to ten (t0 + 6 + 10) days after plastic adherence (t0 + 6). Data expressed as mean ± s.e.m. (average total number of plated EBs/condition/experiment = 48; *$p < 0.05$ ESNO+ vs. ESNO−; $n = 4$). Data analyzed by Kolmogorov-Smirnov test (**d**, **g**) and by 2 way ANOVA (**i**)

**eNOS/NO-positive mESC generate cardiovascular precursors**.
To characterize the mESC subpopulation positive for eNOS
expression and possibly accountable for the cell-autonomous
production of NO (ESNO+), fluorescence-activated cell sorting
(FACS) experiments were performed in the presence of the NO-
activated 4-amino-5-methylamino-2′,7′-difluorescein (DAF)
probe. Figure 6a shows that in GS, the DAF+ subpopulation was
barely detectable (blue color). Instead, Fig. 6b shows the presence
of a DAF+ mESC population quantifiable as early as 2 h after
release from pluripotency. This observation allowed us to sort
cells according to the presence or absence of DAF signal. Of note,
ESNO+ cells, plated for 24 h in DM to allow recovery, did not
show significant morphological differences from ESNO−
(Fig. 6c). After sorting, the expression of miR-200b and mesen-
dodermal markers was analyzed in both cell populations. Simi-
larly to mESC exposed to an exogenous source of NO
(Supplementary Fig. 3a) ESNO+ expressed significantly higher
levels of the NO-inducible miR-200b (Fig. 6d) paralleled by a
decrease in Zeb1 protein level (Fig. 6e). In this condition, Hdac2
was significantly S-nitrosylated compared to ESNO− (Fig. 6f).
Accordingly, mRNA level of *eNOS* and of the other
mesendoderm-associated markers was significantly higher than in
ESNO− (Fig. 6g). Regarding embryoid body (EBs) formation,
ESNO+ and ESNO− populations were both able to generate EBs
(Fig. 6h) although those obtained from ESNO+ grew larger
(Supplementary Table 3). In fact, after plastic adherence (time
point t0 + 6), EBs from both cell types demonstrated the ability to
grow further and differentiate (time point t0 + 6 + 1) (Fig. 6h).
However, in the ESNO+ (open triangles) the number of beating
EBs increased rapidly from time-point t0 + 6 + 3 to time point t0
+ 6 + 10 reaching >60% efficiency in 7 days (Fig. 6i). Conversely,
the formation of detectable beating areas was negligible in EBs
generated by ESNO− (closed squares) (Fig. 6i). qRT-PCR ana-
lysis performed during a time course from t0 to t0 + 6 + 10
indicated that both ESNO+ and ESNO− expressed detectable
and partially overlapping levels of mesendodermal and vascular
markers (Supplementary Fig. 7a). However, *eNOS, Wnt3, Tbx4,
Mesp2*, and *CXCR4* transcripts were significantly higher in ESNO
+ (Supplementary Fig. 7a). Consistently, cardiac-specific mar-
kers, including *Nkx2.5*[45], *Casz1*[46], *Acta1*[47], *Myh6*[48], *Myh7*[48], and
*Cx30.2*[49], were almost exclusively upregulated in ESNO+ (Sup-
plementary Fig. 7b). This result is in agreement with the evidence
that only ESNO+ efficiently formed beating EBs.

## Discussion
NO has always been considered a differentiation agent driving
mESC into mesendodermal/cardiovascular lineage[2–5,7,8]. How-
ever, this gaseous mediator may have contrasting effects[50]. Tejedo
et al. were the first to provide knowledge that, at low doses (2–20
μM), DETA/NO prevented mESC differentiation upregulating
pluripotency genes like *Oct4* and *Nanog*. In that study, similar
results were obtained after expression of wild-type *eNOS* in
undifferentiated mESC[6]. This interesting observation reinforces
the concept that NO donors are pleiotropic agents able to exert
very different effects according to their concentration, as pointed
out by the same authors in other studies[2,51]. Accordingly, data,
showed here in Supplementary Fig. 4a, indicate that mesendo-
dermal transcripts are upregulated in DM, but much more sig-
nificantly in the presence of an external source of NO. Therefore,
a NO donor might provide additional stimuli to mesendodermal
target genes via molecular mechanisms at least in part indepen-
dent from the eNOS, Zeb1, and Hdac1/2 pathway described in
the present manuscript. In spite of these considerations, in our
experiments, the endogenous NO production observed in sorted
cells induced a DAF signal that was similar to that of the unsorted

bulk population treated with DETA/NO. The only difference was
the percentage of positive cells expectedly increasing in the pre-
sence of the NO donor. In parallel, Hdac2 S-nitrosylation
occurred in ESNO+ at a level comparable to that seen in the
bulk population exposed to DETA/NO. Hence, we hypothesized
that, although an exogenous source of NO may have additional
effects, the early production of endogenous NO is sufficient for
introduction of specific PTMs important for early mESC
mesendodermal commitment such as Zeb1 and Hdac2. In
agreement, our findings indicate that the EMT factor Zeb1[21,23]
might actively contribute to self-renewal maintenance keeping
silent important mesendodermal genes including *Mesp2* that with
*Mesp1* is a crucial determinant of early cardiovascular differ-
entiation[37]. Interestingly, prior studies showed that, at later stages
of cardiac morphogenesis, Zeb1 and other transcription factors
involved in EMT, were induced by *Mesp1* and *Mesp2* thus rein-
forcing the evidence of a regulatory developmental feedback loop
among cardiogenic determinants and EMT factors[37]. Along with
this line of evidence, Liu and coworkers[52] recently reported Zeb1
as important for the positive cardiovascular lineage commitment
of human induced pluripotent stem cells (hiPS) or human ESC
(hESC). On the contrary, Kim et al.[53] provided evidence that
Zeb1 downregulation in hESC is an early event during differ-
entiation in endodermal and mesodermal lineages. The latter
evidence is supported by our results and by prior work from Luo
et al[54]. Taken all together these conflicting results may reflect
some degree of heterogeneity in the experimental models adopted
or more likely could be the consequence of the multiple repres-
sion/activation domains present in Zeb family members differ-
entially activated by external stimuli[21].

Zeb1 is a repressive transcription factor that forms complex
with Hdac1 and Hdac2[23,29]. The presence of Hdac2 has been
reported as fundamental to support Zeb1-dependent transcrip-
tion repression properties[23,29]. In our study, we observed that
Zeb1:Hdac2 association was affected by S-nitrosylation since a
source of exogenous NO or its endogenous production increased
Hdac2 S-nitrosylation leading to disruption of the Zeb1:Hdac2
complex. To our surprise, the non-nitrosylatable Hdac2[C262A/
C274A] mutant[12,13] was unable to complex with Zeb1 suggesting
that the same residues might be important for specific protein:
protein association. Cysteines, in fact, are known to play a role in
the formation of intra- and inter-protein complexes[55]. Hence,
Zeb1:Hdac2 interaction might be mediated by Hdac2 cysteines at
position 262 and 274, and their nitrosylation/mutation might
interfere with complex formation and consequently with Zeb1
repression function. Consistent with this evidence, in a Zeb1- or
Hdac2-deprived environment, the expression of mesendodermal
genes was enabled in undifferentiated mESC. This observation
suggested a role for Zeb1:Hdac2 complex in molecular mechan-
isms hindering unscheduled differentiation in mESC. In this
context, the stochastic activation of endogenous NO production
seemed important and associated with eNOS function as indi-
cated by the prevention of differentiation in a genetically engi-
neered environment in which eNOS was knocked-out by
CRISPR/Cas9 technology. NO synthesis abrogation prevented or
significantly reduced expression of mesendodermal markers in
DM. To our knowledge, this is the first time that the endogenous
production of NO, occurring rapidly and spontaneously in mESC,
could be associated with the mesendodermal commitment
orchestrated by a coordinated molecular mechanism requiring
eNOS expression, Hdac2 S-nitrosylation and Zeb1:Hdac2 com-
plex dissociation. In this regard, a body of literature reports that
NO enhances the production of beating EB in mESC[4,5].
Mechanistically this effect remains unclear although some
observations suggest that NO might negatively regulate plur-
ipotency gene expression, like *Nanog*, thus accelerating

differentiation[2]. ESNO+, in fact, isolated according to their NO content, were able to recapitulate all the necessary passages leading to beating EB formation. On the contrary, ESNO−, although viable, were unable to do so. Therefore, ESNO+, in which Hdac2 is S-nitrosylated, might be associated with the formation of cardiac and vascular precursors representing an interesting model to further investigate the early cardiovascuar lineage commitment. Although ESNO− did not express mesendodermal markers they were proficient in transcribing some neuroectodermal-associated genes (Supplementary Fig. 7c). The thorough understanding of the biological and molecular features of the two populations, defined respectively by the presence or absence of eNOS and an active endogenous NO production, requires further analyses.

Although the negative role of Zeb family members in mESC mesendodermal commitment has been reported previously[28,53,54], no molecular mechanisms underpinning its repressive function have been elucidated in this model. Zeb1, as EMT regulator, plays a well-recognized role in tumor progression and metastasization[20,21,23]. In contrast to the ample understanding of the Zeb1 role in human pathophysiology, not much is known about its function during mammalian development. Genetic inactivation studies suggested that Zeb1 is dispensable for mouse gastrulation[24–26], whereas other so-called EMT transcription factors, like Snail, Slug, and Twist, have been indicated fundamental[24]. However, Zeb1 loss-of-function mutation showed a late phenotype characterized by chondrocyte and skeletal defects as well as perinatal lethality due to breathing difficulties of newborn pups[24,26]. In addition, some Zeb1 knock-out animals displayed improper specification of the hematopoietic lineage originating from the mesendodermal germ layer[25]. Notably, the other member of the family, Zeb2, has been identified as an essential regulator of nervous system development[56]. Zeb2 is expressed in the nervous system throughout its development, indicating its importance in neurogenic and gliogenic processes[56]. Indeed, Zeb2 mutations have dramatic neurological consequences both in animal models and in humans determining the Mowat–Wilson syndrome[56]. However, detailed knowledge about Zeb factor role during early embryonic commitment, embryogenesis and tissue homeostasis in postnatal life is lacking. Indeed, Zeb1 and Zeb2 seem to play different roles: Zeb2 more important in neural crest development and central nervous system morphogenesis[56] and Zeb1 involved in neural crest development as well as mesendodermal structure development such as bones, somites, and vessels[24–26]. Hence, Zeb family members have multiple functions, and their inactivation causes complex and partially overlapping phenotypes. The differentiation role of other factors, known to cooperate with Zeb1/2, remains poorly characterized and may have a confounding effect contributing to the complexity of the Zeb-dependent mechanisms. It is not surprising then that the apparent repressive role that Zeb1 exerts during the early mESC mesendodermal differentiation, as seen in this work, is not fully reflecting the phenotype emerging from knock-out experiments which in vivo becomes evident only at much later gestational stages. Nevertheless, the evidence provided here assigns to Zeb1 a role in the negative control of mESC differentiation and embryonic layers commitment, suggesting its function in the maintenance of pluripotency. Zeb1 CRISPR/Cas9 inactivation, in fact, could be associated with loss of undifferentiated state even in cells kept undifferentiated in GS. Other players, including the Krüppel family (Klf) of transcription factors, have been reported to be important for self-renewal maintenance[57,58]. Interestingly, transcription factors, like Klf2 and Klf4, contribute to GS maintenance although they do not participate in mESC differentiation

control but rather in their survival[57]. In this light, Zeb1 might have more selective properties related to the regulation of the early commitment to mesendodermal lineage.

In summary, with this study we experimentally established: (i) the NO-dependent transcription profile in mESC released from the naive state associated with Zeb1 inactivation. (ii) the genome mapping of Zeb1 target genes in the genome of naive mESC. (iii) the isolation of a very early ESNO+ population endogenously synthesizing NO and enriched in mesendodermal precursors committed to cardiovascular differentiation. The identification of the so-called "ESNO+" population might provide a novel model system to investigate early processes associated with the mesendoderm/cardiovascular differentiation. The cartoon depicted in Supplementary Fig. 8 summarizes these concepts and findings.

## Methods

**Cell culture and treatments**. Murine embryonic stem cells (mESC-D3) were purchased from ATCC® and were adapted in culture without a feeder layer. mESC were cultured in standard medium (SM); ground-state medium (GS); spontaneous differentiation medium (DM); and mesendodermal medium achieved adding 250 μM Diethylenetriamine/nitric oxide adduct (DETA/NO-SIGMA-ALDRICH®) as nitric oxide exogenous source (NO). Specifically, SM cultures were performed in DMEM with pyroxidine/HCl (SIGMA-ALDRICH®), supplemented with 1 g L$^{-1}$ glucose, 4 mmol L$^{-1}$ glutamine, 0.1 mmol L$^{-1}$ 2-mercaptoethanol (ROTH), 20 ng mL$^{-1}$ recombinant mouse leukemia inhibitory factor (LIF-Millipore) and 10% FBS pre-tested for ES cells (PAA). To achieve GS, mESC were cultured in NDiff N2B27 media (Stem Cells, Inc.; scs-sf-nb-02) in the presence of the following three inhibitors (2i/L): 1 μM PD0325901 MEK inhibitor (SIGMA-ALDRICH®), 3 μM CHIR99021 GSK-3 inhibitor XVI (Millipore) and 20 ng mL$^{-1}$ LIF (Millipore). DM cultures were performed in DMEM with pyroxidine/HCl (SIGMA-ALDRICH®), supplemented with 1 g L$^{-1}$ glucose, 4 mmol L$^{-1}$ glutamine, 0.1 mmol L$^{-1}$ 2-mercaptoethanol (ROTH) and 10% FBS Pre-tested for ES cells (PAA). NO cultures were performed in DMEM with pyroxidine/HCl (SIGMA-ALDRICH®), supplemented with 1 g L$^{-1}$ glucose, 4 mmol L$^{-1}$ glutamine, 0.1 mmol L$^{-1}$ 2-mercaptoethanol (ROTH), 10% FBS pre-tested for ES cells (PAA) and 250 μM diethylentriamine/nitric oxide adduct (DETA/NO-SIGMA-ALDRICH®). mESC were treated with the following chemicals: 250 μM DETA/NO (SIGMA-ALDRICH®), 100 μM NO scavenger 2-phenyl-4,4,5,5-tetramethylimidazoline-1-oxyl 3-oxide (PTIO-SIGMA-ALDRICH®) or 1 μM 1H-[1,2,4]oxadiazolo[4,3-a] quinoxalin-1-one (ODQ-Cayman chemical) and 5 mM N5-[imino(nitroamino) methyl]-ʟ-ornithine, methyl ester, monohydrochloride (L-NAME-Cayman chemical).

Human cervical carcinoma HeLa cells were purchased from ATCC® and were cultured in DMEM (SIGMA-ALDRICH®), supplemented with 4 mmol L$^{-1}$ glutamine, 1% Penicillin–Streptomycin (SIGMA-ALDRICH®) and 10% FBS (Gibco). 1 μg pCIG2_Hdac2$^{wt}$, pCIG2_Hdac2$^{C262A/C274A}$[12,13] or empty vector were transfected in HeLa cells using Lipofectamine 3000 (Invitrogen) according to the manufacturer's instructions. Three days later cells were collected for co-immunoprecipitation experiments.

**Western blot and immunoprecipitation**. Western blotting was performed by standard procedures after cell lysis in Laemmli buffer. Nitrocellulose blotting membranes were probed with the following antibodies: Zeb1 (Santa Cruz; 1:500); Klf4 (Abcam; 1:1000); Nanog (Abcam; 1:1000); Oct4 (Abcam; 1:800); Sox2 (Abcam; 1:1000); p53 (Santa Cruz; 1:1000); Acetylated p53 (Cell Signaling; 1:1000); Hdac2 (Santa Cruz; 1:1000); eNOS (BD; 1:1000); iNOS (Cell Signaling; 1:1000); nNOS (Cell Signaling; 1:1000); cMyc (Cell Signalling; 1:2000); fibrillarin (Santa Cruz; 1:1000); histone 3 (H3-Abcam; 1:1000); α-tubulin (Cell Signaling; 1:5000); and β-actin (SIGMA-ALDRICH®; 1:4000). Development was performed by Odyssey CLX reader (LI-COR). Coimmunoprecipitations were performed using 500 μg extracts after lysis of samples in 50 mM Tris-HCl (pH 7.4), 150 mM NaCl, 1% Triton X-100, 2 mM MgCl$_2$ and 1% sodium deoxycholate supplemented with 1 mM PMSF and protease inhibitor mix using 4 μg of anti-S-nitrosocysteine (Alpha Diagnostic), 4 μg of anti-Hdac2 (Santa Cruz) or 10 μl of anti-Zeb1 (Active motif). The Ademtech Bioadembeads paramagnetic beads system was used to immunoprecipitate the specific proteins according to the manufacturer's instructions. Full-length images of all representative western blots related to the present manuscript are provided as Supplementary Fig. 9, 10, 11, 12, 13, 14, and 15. The list of all used antibodies is provided in Supplementary Table 4.

**Subcellular fractionation**. 5 × 10$^6$ mESCs from GS, DM, and NO conditions were lysed with the lysis buffer provided by Qproteome Cell Compartment kit (QIAGEN), then fractionation was performed according to the manufacturer's instructions. Subcellular fraction content was normalized according to Comassie staining before western blotting.

**mRNA extraction and qRT-PCR**. RNA was extracted from mESC (approximately 10⁶) using Tri-Reagent (SIGMA-ALDRICH®) according to the manufacturer's instructions. cDNA synthesis for quantitative real-time PCR (qRT-PCR) was carried out with SuperScript III First-Strand Synthesis Super Mix for qRT-PCR (Invitrogen) according to the manufacturer's protocol. All reactions were performed in 96-well format in the StepOne Plus Real-Time PCR System (Applied Biosystems) using PerfeCTa® SYBRGreen® FastMix®, ROX™ (Quanta BIOS-CIENCES™). For each gene of interest, qRT-PCR was performed as follows: each RNA sample was tested in duplicate and p0 was used to normalize transcript abundance. mRNA expression levels were calculated by Comparative Ct Method by using the Applied Biosystem software (Applied Biosystem, CA, USA) and were presented as fold induction of transcripts for target genes. Fold change above 1 denotes upregulated expression, and fold change below 1 denotes downregulated expression vs. cells cultured in GS condition.

List of forward and reverse primers is provided in Supplementary Table 5.

The sequences were selected based on published sequence data from NCBI database. Primers for miR-200c, miR-200a, miR-200b, miR-429, miR-141, miR-16, primiR-200 cluster1, primiR-200 cluster2, and the reagents for reverse transcriptase and qPCR reactions were all obtained from Applied Biosystems. miRNA expression levels in each sample were normalized to miR-16 expression as, under the experimental conditions of the present study, miR-16 was not modulated by mESC differentiation or treatment.

**RNA sequencing and bioinformatics analysis**. Next-generation sequencing was performed on Ion Torrent Proton sequencing platform (Thermo Fisher) using the Ion Total RNA-Seq Kit v2 (Thermo Fisher) with minor modifications. Total RNA for transcriptome analysis was isolated with miRNeasy Micro Kit (Qiagen) from a 60 mm² dish per sample. The integrity of RNA was checked on Bioanalyzer 2100 (Agilent), and 5 μg of RNA with RIN >9 were used for ribosomal depletion using the Ribo-Minus Eukaryote Kit v2 (Invitrogen). Following modifications to the standard RNA-seq Library protocol were performed: fragmented RNA was concentrated to 3 μL with SpeedVac (Eppendorf) for 10–15 min, and all 3 μL were used for further steps. The number of amplification cycles was reduced to 8–12 cycles resulting in lower PCR duplication levels without any influence to quality/contribution of obtained reads. To maximize data, recovery libraries were size selected using the LabChipXT system (Perkin Elmer) with an isolation window between 180 and 270 bp (corresponding to 60–150 bp insert size). Obtained RNA libraries were quantified on Qubit 2.0 and diluted to 100 pM and used in a final concentration of 10 pM for template preparation on Ion OneTouch2 instrument (Thermo Fisher). For each run on the PI Ion Torrent V2 Chip, 2 RNA libraries were pooled in equimolar ratios to obtain between 37 and 63 M raw reads per sample. The resulting raw reads were assessed for quality, adapter content and duplication rates with FastQC (Andrews S. 2010, FastQC: a quality control tool for high throughput sequence data. Available online at [http://www.bioinformatics.babraham.ac.uk/projects/fastqc]). Reaper version 13–100 was employed to trim reads after a quality drop below a mean of Q20 in a window of 10 nucleotides[59]. Only reads between 30 and 150 nucleotides were cleared for further analyses. Trimmed and filtered reads were aligned vs. the Ensembl mouse genome version mm10 (GRCm38) using STAR 2.4.0a with the parameter "--outFilterMismatchNoverLmax 0.1" to increase the maximum ratio of mismatches to mapped length to 10%[60]. The number of reads aligning to genes was counted with featureCounts 1.4.5-p1 tool from the Subread package[61]. Only reads mapping at least partially inside exons were admitted and aggregated per gene (see Supplementary Table 1). Reads overlapping multiple genes or aligning to multiple regions were excluded. Differentially expressed genes were identified using DESeq2 version 1.62[62]. Only genes with a minimum fold change of ±2, a maximum Benjamini–Hochberg corrected p-value of 0.05, and a minimum combined mean of 5 reads were deemed to be significantly differentially expressed. The Ensemble annotation was enriched with UniProt data (release 06.06.2014) based on Ensembl gene identifiers (Activities at the Universal Protein Resource (UniProt)). The correlation of replicate gene counts was assessed with the Spearman ranked correlation algorithm included in R 3.11 (R: A language and environment for statistical computing). MA plots were computed using the script run_DE_analysis.pl included in Trinity version 20140717 which employs R functions for plotting[63]. After pairwise comparison of GS/DM, NO/GS, and NO/DM the overlapping extent of differentially expressed genes among experimental conditions was analyzed in up- and downregulated RNAs (±2 log2 fold change, basemean >5, fdr <0.05) by Venn diagrams (Venny 2.1.0 [http://bioinfogp.cnb.csic.es/tools/venny/]). Gene ontology on genes exclusively regulated by NO (±2 log2 fold change, basemean >5, FDR <0.05) was performed using Cytoscape 3.2.1 plugin BINGO, which allows to determine statistically overrepresented GO categories in the derived biological networks.

**Chromatin immunoprecipitation**. mESCs were cultured in GS, after inhibitor withdrawal (DM) and after inhibitor withdrawal plus supplementation of 250 μM DETA/NO (NO) (SIGMA ALDRICH®). At 24 h, cross-linking with 1% formaldehyde (Applichem) and quenching with 10× Glycine (Millipore) as well as nuclear extraction were performed. Then, chromatin solution was sheared by Bioruptor® Plus (Diagenode) to obtain about 500 bp fragments for qRT-PCR detection or 200 bp fragments for Sequencing. Immunoprecipitation protocol was

performed after overnight incubation at 4 °C with 1 μg of p53 (Pab240 from Santa Cruz), 8 μg of Zeb1 (H-102 from Santa Cruz), 10 μg of Hdac2 (GeneTex) or 4 μg of Hdac1 (Abcam) antibody with Magna-ChIP™ A/G (Millipore) kit according to manufacturer's instruction. Details related to used antibodies are provided in Supplementary Table 4. DNA fragments were recovered and analyzed either by Sequencing or by qRT-PCR (0.5–1 μL of immunoprecipitated chromatin) using the primers listed in Supplementary Table 6.

The qRT-PCR analyses were performed in duplicate and the data obtained were normalized to the corresponding DNA input control. Data are represented as relative enrichment (with values for IgG being subtracted from those with antibody).

**ChIP sequencing and bioinformatics analysis**. ChIP samples were processed for sequencing on the Ion Torrent platform using a modified DNA Library protocol based on the Ion Xpress Plus Fragment Library Kit (Thermo Fisher). 1–10 ng dsDNA were used as starting material. After end repair of DNA fragments, Ion Torrent specific and barcoded DNA, Adapters were ligated and covalently bound by nick translation, followed by amplification with 18 cycles using the components of the Ion Xpress Plus Fragment Library Kit. The final SPRI bead cleanup was performed as double size selection with 0.7× DNA/Bead ratio followed by 0.8× DNA/bead ratio. QC of obtained libraries was done on Bioanalyzer 2100 (Agilent) and quantitation was measured on Qubit 2.0 with HS DNA Assay. Libraries were diluted to a final concentration of 10 pM and used in equimolar ratios for template preparation on Ion Touch 2 instrument. Libraries were pooled on PI Ion Torrent V2 Chips to obtain a minimum amount of 20 M raw reads per sample. The quality of raw reads was assessed using FastQC (see Andrews S. 2010, FastQC: a quality control tool for high throughput sequence data. Available online at [http://www.bioinformatics.babraham.ac.uk/projects/fastqc]). Reads were trimmed for a minimum average quality of Q20 in a window of 20 nucleotides using Reaper 13.274 from the EBI Kraken package[59]. Only reads between 20 and 150 nucleotides were cleared for further analysis. To normalize all samples to the same sequencing depth, 30 million reads per sample were randomly selected for further analysis. These were mapped onto mm10 (GRCm38) version of the mouse genome with STAR 2.4.2a[60] using only unique alignments to exclude reads with unclear placing (--outFilterMismatchNoverLmax 0.1 --alignIntronMin 2 --alignIntronMax 1 -- alignEndsType EndToEnd --outFilterMultimapNmax 1). The reads were further deduplicated using Picard 1.136 (see Picard: A set of tools (in Java) for working with next-generation sequencing data in the BAM format; [http://broadinstitute.github.io/picard/]) to avoid PCR artifacts leading to multiple copies of the same original fragment (see Supplementary Table 1). The ENCODE peak caller PeakRanger 1.18[64] was employed in ccat-mode to accommodate for broad peaks. PeakRanger automatically normalizes input and treatment reads and calls peaks taking the background distribution into account. To determine significant peaks, the data were manually inspected in IGV 2.3.52 to identify reasonable thresholds[65]. Significant peaks were deemed to show at least 20 treatment reads, a minimum enrichment of treatment versus input reads of 2-fold, and a maximum FDR of 0.05. Peaks overlapping ENCODE blacklisted regions (known misassemblies, satellite repeats) were excluded. Only those peaks that were present in at least two replicates (overlap ≥1 nucleotide) were permitted for further analysis. To be able to compare peaks in different samples to assess reproducibility, the resulting three lists of significant peaks (one per treatment replicate) were overlapped and unified to represent identical regions. Input and treatment read coverage were recomputed for the unified peaks for each sample and normalized to the same depth using factor normalization. For each peak, input reads were subtracted from treatment reads to correct for mapping irregularities. The background corrected counts were then submitted to DESeq[66] for further normalization considering differences of sequencing depth between samples and outlier correction. For the final peak list, values of unified peaks per sample (treatment reads, input reads, and enrichment) were averaged. Unified peaks were annotated with Homer 4.7[67] based on data from RefSeq mm10. Peaks were determined to overlap a promoter if their center was located less than 5 kb up- or downstream from the TSS of the respective gene. Profiles for the peak overlap of e.g. TSS or gene bodies were computed using CEAS 1.0.2[68] using default parameters. The correlation of replicate peak counts was assessed with the Spearman ranked correlation algorithm included in R 3.11 (R: A language and environment for statistical computing). Intersection between RNASeq and ChIPSeq (FDR ≤ 0.05; enrichment = 20; treatment reads 100–5000) was determined by Venn diagrams (Venny 2.1.0 [http://bioinfogp.cnb.csic.es/tools/venny/]). Gene ontology on genes belonging to the intersection list was performed using Cytoscape 3.2.1 plugin BINGO, which allows to determine statistically overrepresented GO categories in the derived biological networks.

**cGMP ELISA quantification**. mESC lysates were assessed in GS condition, at 1, 2, and 3 h time points after inhibitor withdrawal (DM) in absence or presence of 100 μM nitric oxide scavenger 2-phenyl-4,4,5,5-tetramethylimidazoline-1-oxyl 3-oxide (PTIO) (SIGMA-ALDRICH®) or 1 μM 1H-[1,2,4]oxadiazolo[4,3-a]quinoxalin-1-one (ODQ-Cayman chemical) or 5 mM N5-[imino(nitroamino)methyl]-L-ornithine, methyl ester, monohydrochloride (L-NAME-Cayman chemical) and at 24 h after nitric oxide treatment with 250 μM DETA/NO (NO) (SIGMA-ALDRICH®) ±100 μM PTIO. Total protein extract concentration was quantified by BCA Assay (Pierce). cGMP determination was performed by cGMP ELISA Kit (Colorimetric-

Cell Biolabs, Inc.) according to the manufacturer's instructions using 50 µg of protein extract in duplicate.

**Confocal microscopy**. Confocal analysis was performed as reported previously[69]. Zeb1 (Santa Cruz; 1:150) was used, and nuclei were counterstained with DAPI solution (1:5000). Samples were analyzed using a Leica TCS SP8 confocal microscope.

**FACS analysis**. Nitric oxide production was evaluated by adding 4,5-diamino-fluorescein diacetate (DAF-2DA, Cayman Chemical; 1:2000) according to the manufacturer's instructions to mESC cultured 1, 2, 3, 6, 24 h in DM or NO. At the end of treatment, cells were collected and analyzed by FACS (FACS Canto II-BD) to detect intracellular NO production.

**Cell sorting**. mESCs were cultured in GS condition and 3 h inhibitor withdrawal condition (DM) in the presence of an NO fluorescent probe 4,5-Diamino-fluorescein diacetate (DAF-2DA, Cayman Chemical; 1:2000) according to the manufacturer's instructions. Then, cells were detached by Triple Express Enzyme w/o Phenol Red (Life Technologies) and resuspended in DMEM with pyroxidine/HCl (SIGMA-ALDRICH®), supplemented with $1 g L^{-1}$ glucose, $4 mmol L^{-1}$ glutamine, $0.1 mmol L^{-1}$ 2-mercaptoethanol (ROTH) and 10% FBS Pre-tested for ES cells (PAA). Sorting setup and appropriate gating were established each time using mESC cultured in DM in the absence of DAF-2DA. To minimize cell death and maximize cell recovery sorting was performed by using FACS ARIA III from BD (Beckton Dickinson).

**Embryoid body assay**. After sorting mESC were cultivated and differentiated to the cardiac lineage by a hanging drop technique (HD). In brief, six days after formation (t0 + 6), EBs were transferred into a 24-well 0.1% gelatin-coated multiwell. Total differentiation time is indicated as 6 + N, where 6 indicates the time in suspension and N is the time after plating. Beating areas within plated EBs ($n = 48$/condition) were counted daily and used as indication of cardiomyocyte differentiation. The portions of the contractile EB have been identified under the microscope by phase contrast and isolated mechanically using a sterile disposable scalpel. For each experiment were harvested contractile areas of about 5 EBs and then analyzed for mRNA expression of mesendodermal, endothelial, neuroectoderm, and cardiac markers.

**CRISPR/Cas9 vector generation**. To knock-out Zeb1, Hdac2, and eNOS the target-specific sgRNAs, listed in Supplementary Table 7, were cloned into Lenti-CRISPR2 vector (addgene, Cambridge, MA) using the GoldenGate protocol[70].

All CRISPR/Cas9 experiments were compared to non-targeting control (NTC) obtained with the sgRNAs, listed in Supplementary Table 7, cloned into the LentiCRISPR2 vector using the GoldenGate protocol.

The obtained plasmids were transformed into NEB 5-alpha Competent *Escherichia coli* (High Efficiency –New England Biolabs), then DNA was purified by EZNA Fastfilter Endo-Free Plasmid DNA Maxi Kit (Omega Bio-Tek), and a concentration of 6 µg was used for nucleofection. Nucleofection was performed in $10^6$ mESC cultured in GS using Amaxa P3 primary cell 4D Nucleofector Kit (Lonza). After 48 h, nucleofected mESC were selected by $1.5 µg mL^{-1}$ puromycin. After recovery from selection, mESC were tested for Zeb1, Hdac2, and eNOS knockout by western blot. mESC nucleofected with eNOS CRISPR/Cas9 vectors resulted knocked out and were used for subsequent experiments, whereas mESC nucleofected either with Zeb1 CRISPR/Cas9 vectors or Hdac2 CRISPR/Cas9 vectors were clonally expanded. Once monoclonal Zeb1_1 and Zeb1_2 CRISPR/Cas9 mESC as well as monoclonal Hdac2_1 and Hdac2_2_CRISPR/Cas9 mESC were obtained, expression analysis of mesendodermal markers was conducted.

**Adenovirus generation and adenoviral transduction of cells**. Replication-deficient adenoviruses for the expression of $eNOS^{wt}$, $eNOS^{S1177D}$, or GFP were generated and expanded using the AdEasy Adenoviral Vector System (Agilent Technologies). Briefly, pShuttle-CMV vectors carrying $eNOS^{wt}$, or $eNOS^{S1177D}$ and pAdTrack-CMV were linearized with Pme I and subsequently cotransformed into *E. coli* BJ5183 cells with an adenoviral backbone plasmid, pAdEasy-1. Recombinants were selected by kanamycin resistance and verified by restriction enzyme digestion. The confirmed recombinant plasmids were then transfected into the adenoviral packaging AD-293 cell line. Viral production was monitored over 7–10 days by visualization of GFP expression and cytopathic effect (CPE). After 7–10 days, viral particles were harvested and purified using the AdenoONE™ Purification Kit (Sirion Biotech). The titer of the purified adenoviruses was determined by standard plaque forming unit assay. Viral stocks were maintained at −80 °C for long-term storage before use. mESC were infected with 50 multiplicities of infection (MOI) of each adenovirus after 30 min incubation with Adeno-BOOST™ Adenovirus Transduction Enhancer Solution (Sirion Biotech), following the manufacturer's instructions. Infected mESC were cultured for additional 48 h, treated in GS, DM or NO for 24 h and then collected for subsequent experiments.

**Lentiviral infection**. Lentiviral supernatants were purchased from OriGene. mESC were infected for 16 h with lentiviral particles expressing GFP (Lenti_Empty) or Zeb1 (Lenti_Zeb1-OriGene), then mESC were allowed to recover in complete fresh medium for additional 48 h. Afterwards, infected mESC were treated in GS, DM or NO for 24 h and then collected.

**miR200b-LNA nucleofection**. Nucleofection was performed in $10^6$ mESC cultured in GS using Amaxa P3 primary cell 4D Nucleofector Kit (Lonza) either with 50 µM Mircury scramble or miR-200b LNA-oligonucleotides (Exiqon). After 16 h, cells were incubated with fresh medium for 32 h and then treated in GS, DM or NO for additional 24 h.

**Statistical analysis**. Statistical analyses were performed using GraphPad Prism programme. Sample sizes ($n$) were reported in the corresponding figure legend. No statistical method was used to predetermine sample size. None of the samples was excluded from the experiments. Investigators performing sequencing analysis, ChIP-qRT-PCR and immunoprecipitation were blinded during the experiment. All values were presented as mean ± the standard error of the mean (s.e.m.) or standard deviation (s.d.) of at least three independent experiments, unless otherwise indicated. Statistical analyses were performed using non-parametric Student's $t$-test and two-way ANOVA. For all statistical analysis, a value of $p \leq 0.05$ was deemed statistically significant.

**Data availability**. All relevant data are available from the authors. The RNA and DNA sequencing datasets generated and analyzed during the current study are available in the GEO repository [http://www.ncbi.nlm.nih.gov/projects/geo/query/acc.cgi?acc=GSE104649].

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

## Acknowledgements

The present study was supported by: LOEWE Cell & Gene Therapy Center (LOEWE-CGT) Goethe University Frankfurt and by Deutsche Forschungsgemeinschaft Program SFB834 "Endothelial Signaling and Vascular Repair", Project B11 grants to C.G. C.C. is recipient of the Start up grant 2016 from LOEWE-Forschungszentrum für Zell- und Gentherapie, gefördert durch das Hessische Ministerium für Wissenschaft und Kunst.

Aktenzeichen: III L 5–518/17.004 (2013) and funded by August Scheidel-Stiftung and Amandus und Barbara Pauli Stiftung 2016. F.S. is recipient of the LOEWE CGT grant # III L 5–518/17.004 (2013) and funded by the DFG (German Research Foundation) Excellence Cluster Cardio Pulmonary System. A.R. is supported by a fellowship of the Italian Association for Cancer Research (AIRC) and by the CNR Short-Term Mobility Program 2016. This work was supported by Ministero della Salute (Ricerca Corrente, 5X1000, RF-2011-02347907 and PE-2011-02348537), Telethon-Italy (grant#GGP14092), and AFM Telethon (grant #18477) to F.M. We thank Lena Dorsheimer for her help with sorting during the second revision of the present paper.

## Author contributions

C.C. and F.S. conceived and carried out experiments and data analysis and revised the manuscript; C.K. designed and carried out the bioinformatics analysis; M.S., S.G.,. A.R., S.W., M.R., D.S., M.S., J.G.A. carried out the experiments; F.Sc., H.v.M., M.A.R., F.M., A. R., I.F., A.F., T.B., A.M.Z. revised data analysis and the manuscript; C.G. conceived the experiments and wrote the manuscript. All the authors contributed critical discussion and approved the final version of the manuscript.

## Additional information

**Competing interests:** The authors declare no competing interests.

