## [Peer Review File(PDF 2117 kb) · Nature Communications]

Editorial Note: Several images have been redacted to protect copyright. The images can be found in the original publications, cited at the end of the figure legends.

Reviewers' comments:

Reviewer #1 (Remarks to the Author):

I am generally comfortable with the novel results and conclusion that eNOS/HDAC/Zeb1 (through destabilization of the Zeb1/chromatin interaction) triggers mesoendodermal cardiovascular commitment. Integrated RNA-seq/Chip-seq point to the role of NO/Zeb1 in mesodermal differentiation. Expression patterns of eNOS vs. n/iNOS point to the role for eNOS after release from stemness. eNOS knockdown blocks mesoendoderm-associated genes. HDAC2/Zeb knockdown influenced eNOS expression.

I suggest a few control experiments to strengthen the mechanistic aspects of the paper.

1. Please employ an nNOS inhibitor to exclude complementary effects of nNOS.
2. What effect does a pan NOS inhibitor have cardiovascular commitment?
3. What effect does ODQ have?
4. Mechanistic studies on HDAC nitrosylation are mostly correlative. Please overexpress an HDAC2 Cys mutant in cells deficient in HDAC to confirm causality.

Reviewer #2 (Remarks to the Author):

This manuscript analyzes the transcription profile of nitric oxide (NO) and of Zeb1 in mouse stem (mES) cells. Cencioni et al claim that their work identified NO-dependent Zeb1 targets and that they isolated an early mES population that synthesizes NO and is enriched in mesodermal precursors committed to cardiovascular differentiation. The authors also conclude that their results support a role for Zeb1 as repressor of mesodermal differentiation and that NO, through its effect on the S-nitrosylation of HDAC2, overcomes Zeb1 repressor action.

General comments

The work suffers from a number of important shortcomings detailed below. But the main criticism is that the conclusions raised by the authors are not supported by the experimental data. In fact, the authors have not performed the experiments required to reach those claims (see below). For instance, one of the major conclusions of the paper (included in the title) is that Zeb1 identifies and regulates key genes in ES cells. As discussed below, the authors failed to obtain an efficient CRISP / Cas9 targeting of Zeb1. Even more surprisingly, the resulting targeted cells were not used to study the role of Zeb1 in differentiation or in the regulation of the selected genes. In addition, none of those genes were validated for direct Zeb1 regulation. The manuscript also overlooks several publications directly related to the topic.

Lastly, the manuscript contains numerous mistakes in the description of figures, numbering/labeling of panels, definition of culture conditions, etc. While these issues are

usually considered "minor comments," they are not trivial in this work, particularly considering the importance of well-defined culture conditions in ES work.

Main criticisms

1) The study deals with an interesting topic for which there is a number of published articles. The work follows up on a previous paper by the same authors (that surprisingly it is not cited) in Spallotta et al (2010) *Stem Cells*, 28:431-42. The manuscript also overlooks two articles directly related to the topic, namely: Mora-Castilla et al (2010). *Cell Death & Diff*, 17:1025-1033 and Tejedo et al (2010). *Cell Death & Disease* 1:e80. In their discussion of Mesp2 and Zeb1, authors should also cite and discuss *J Cell Biol* (2016) 4:463-7. There is also evidence in the literature that Zeb1 is not regulated by NO in epithelial but these works are not cited either.

2) The definition of the media and culture conditions is confusing. In addition, in several experiments, some culture conditions are not included without explanation (see points 3 and 4). Obviously, this is a key issue when working with ES. For instance in line 66, authors indicated that they use FCS but in Methods they talk about FBS. In other example, two different culture conditions are labeled with the same name. In line 72-73: "in response to LIF withdrawal (DMSMNO), an effect further enhanced in the presence of NO (DMSMNO)". One has to assume that it is meant DMSM but this mistake only adds to the confusion.

3) In Supplemental Figure 1, the authors show the differences between SM and GS cultures with regard to NO production. However, they do not measure NO levels before NONOate administration as well as after 24hrs of NONOate administration.

4) In Supplemental Figure 2a, the levels of Zeb1 under SM culture conditions should be shown.

5) Authors claimed that in Supplemental Figure 2, panel b, DMGSNO "Zeb1 clearly relocate out the nucleus". Despite the authors claim that Zeb1 "clearly" relocates, I am not convinced by the data shown. At best, this figure only confirms the decrease in Zeb1 expression shown in "panel a" but the relocation of Zeb1 is not convincing. Most cells in the DMGSNO condition express little Zeb1 (if any at all). In sum, their data do not allow to conclude that Zeb1 is translocated. Such claim should be backed up by a western blot of nuclear and cytoplasmic fractions.

6) Figure 3, panel a. It is not clear why the authors do not show the DMGSNO condition at 1 h and 2h as they show these time points for the other culture conditions. The explanation of this Figure in the main text (line 163) indicates "gray stripped bars" but there are not gray stripped bars in Figure 3, panel a.

7) The efficiency of CRISP/Cas9 targeting of Zeb1 in Supplemental Fig 4c is very poor and questions the conclusions that authors make out of the targeting experiments. It would be

essential to obtain better deletion of Zeb1 (see also below).

8) Supplemental Fig 3b shows that Zeb1 was enriched in selected genes in the GS culture condition. Surprisingly, none of these genes is examined with the CRISP/Cas 9 targeted cells (LCv2_Zeb1_1 and LCv2_Zeb1_2). In addition, regulation of these genes by Zeb1 should be validated by quantitative PCR.

9) Very importantly, and key to justify their conclusions: the authors should analyze the differentiation genes in Figures 3 panel g and Figure 5 using the cells CRISP/Cas9 targeted for Zeb1 and Hdac2 and under the different culture conditions (GS, DMGS, DMGSNO). Likewise, they should examine the expression of these genes in the cells targeted for Zeb1 and Hdac2 with overexpression of eNOs. Without those experiments, it is difficult to claim that Zeb1 is regulating these genes, let alone including such claim in the abstract and title of the manuscript.

10) Authors should both characterize Zeb1 expression and effect of Zeb1 elimination in eSNO- and eSNO+ cells.

11) It should be demonstrated that the elimination of Zeb1 in eSNO- cells increases differentiation and miR-200 levels. In addition, miR-200 should be knockdown in eSNO+ cells to inhibit differentiation

12) Zeb1 should be overexpressed and examine the number of beating EBs formed

13) My understanding is that Zeb1 knock out mice does not have defects in cardiovascular development (Takagi et al., Development 125, 21-31,1998). Authors should at least discuss how their results fit with the phenotype of these mice. In light of the phenotype of the Zeb2 knock out mice, I feel it is unlikely that this could be explained either by compensation between Zeb1 and Zeb2

Other comments

1) Several figures and panels are mislabeled. This is particularly the case of Figure 4. The panels explained and referred in the main text (lines 215-219) do not correspond to the panels shown in Figure 4.

2) While Supplemental Figures are numbered and labeled, the main Figures (at least the version that this Reviewer received) are not

3) There is no information about the identity of the Ab used in the Zeb1 ChIP-seq, beyond that is a Santa Cruz antibody. This is a highly relevant piece of information considering that the quality of the Ab used is essential in ChIP-seq analysis.

4) Line 253. The correct name of the GSK3 inhibitor is CHIR, not CHYR.

6) Mouse gene HDAC2 should be written as Hdac2

Reviewer #1

I am generally comfortable with the novel results and conclusion that eNOS/HDAC/Zeb1 (through destabilization of the Zeb1/chromatin interaction) triggers mesoendodermal cardiovascular commitment. Integrated RNA-seq/Chip-seq point to the role of NO/Zeb1 in mesodermal differentiation. Expression patterns of eNOS vs. n/iNOS point to the role for eNOS after release from stemness. eNOS knockdown blocks mesoendoderm-associated genes. HDAC2/Zeb knockdown influenced eNOS expression.

I suggest a few control experiments to strengthen the mechanistic aspects of the paper.

The authors would like to thank the Reviewer for His/Her positive comments to their work.

Q1. Please employ an nNOS inhibitor to exclude complementary effects of nNOS.

A1. *The authors thank the Reviewer for the important suggestion. However, in light of the evidence that the specific nNOS inhibitor Nω-propyl-L-Arginine has been reported as not fully selective for nNOS at least in an in vivo context (Gowda C, Toomayan GA, Qi WN, Chen LE, Cai Y, Allen DM, Seaber AV, Urbaniak JR. The effects of N(omega)-propyl-L-arginine on reperfusion injury of skeletal muscle. Nitric Oxide. 2004 Aug;11(1):17-24), we decided to address the question about the contribution of nNOS in the early NO production of ES further analysing mES where endogenous eNOS was inactivated by CRISPR/Cas9 technology. These experiments indicate that the most important effector of endogenous NO synthesis is actually eNOS. Its inactivation, in fact, determined a significant drop in NO synthesis after release from stemness in spite of a well detectable presence of nNOS. The new data have been now provided respectively in figure 5, panel a and suppl. fig. 4, panel d of the revised manuscript version.*

Q2. What effect does a pan NOS inhibitor have cardiovascular commitment?

A2. *The pan-inhibitor L-NAME has been used to address this question. Data indicate that in the presence of this pan-inhibitor the production of cGMP is abrogated in early released ES cells and the expression of eNOS, Wnt3, Dll1 and Wnt7b is significantly reduced. These results are now depicted respectively in figure 3 panel a and suppl. Fig. 4 panel c.*

Q3. What effect does ODQ have?

A3. *The GC inhibitor ODQ has been used to monitor the early ES production of NO. Figure 3 panel a shows that ODQ prevents cGMP synthesis in ES released from stemness. However expression analysis shown in suppl. Fig. 4, panel c indicates that ODQ does not interfere with mesendodermal marker expression. These results suggest that cGMP synthesis may not be crucial for the NO downstream effect on Zeb1 target gene expression. A specific comment has been reported in the Results section of the revised manuscript at page 11 in the paragraph entitled "eNOS is important for mesendodermal gene expression".*

Q4. Mechanistic studies on HDAC nitrosylation are mostly correlative. Please overexpress an HDAC2 Cys mutant in cells deficient in HDAC to confirm causality.

A4. *The authors would like to thank Reviewer for the important suggestion. New experiments have been performed in the attempt of providing a role for the Hdac2 cys mutant. The results indicate that the double mutant Hdac2^{C262A/C274A} is poorly associated to Zeb1 (figure 3, panel e), thus reproducing the effect of nitrosylation. This result suggests that the cysteines modified by nitrosylation are also important for Hdac2:Zeb1 interaction. However the evidence that in absence of the specific cysteines or in the presence of nitrosylation Zeb1:Hdac2 association decreases prevented us from performing further analysis.*

Reviewer #2

The authors would like to thank the Reviewer for His&Her useful comments. The following are their point/to/point answers to Reviewer questions.

This manuscript analyzes the transcription profile of nitric oxide (NO) and of Zeb1 in mouse stem (mES) cells. Cencioni et al claim that their work identified NO-dependent Zeb1 targets and that they isolated an early mES population that synthesizes NO and is enriched in mesodermal precursors committed to cardiovascular differentiation. The authors also conclude that their results support a role for Zeb1 as repressor of mesodermal differentiation and that NO, through its effect on the S-nitrosylation of HDAC2, overcomes Zeb1 repressor action.

Q1. *The work suffers from a number of important shortcomings detailed below. But the main criticism is that the conclusions raised by the authors are not supported by the experimental data. In fact, the authors have not performed the experiments required to reach those claims (see below). For instance, one of the major conclusions of the paper (included in the title) is that Zeb1 identifies and regulates key genes in ES cells.*

A1. *The authors would like to thank the Reviewer for giving us the possibility to clarify our findings. Specifically, figure 2, panel b clearly shows the presence of Zeb1 binding sites (blue color peaks) on mesendodermal genes identified by intersecting RNA-Seq and ChIP-Seq (figure 2, panel a): namely Wnt3, Wnt7b, Dll1, Tbx4 and Mesp2. Moreover, suppl. fig. 3, panel b shows the results of VALIDATION experiments performed by ChIP/qRT-PCR analysis. These experiments added further information about the presence of Zeb1 binding sites on those genes: Zeb1 binding is significantly weakened by the release of mES from ground state into differentiation.*

Q2. *As discussed below, the authors failed to obtain an efficient CRISP / Cas9 targeting of Zeb1. Even more surprisingly, the resulting targeted cells were not used to study the role of Zeb1 in differentiation or in the regulation of the selected genes. In addition, none of those genes were validated for direct Zeb1 regulation.*

A2. *We agree that CRISPR/Cas9 targeting of Zeb1 was apparently less efficient than that of Hdac2 and eNOS. However, this is the consequence of analyses performed in uncloned bulk cell populations. Nevertheless, we would like to raise the Reviewer attention on the fact that two independent targeting constructs were always used in our experiments and, most importantly, on the fact that a clear phenotype emerged in spite of the apparent partial Zeb1 targeting. However, in the revised manuscript version new experiments were performed to address this specific point. Different clones of Zeb1 CRISPR/Cas9 mES have been isolated and analysed (suppl. fig. 4, panel a left). The results are shown in the new figure 4, panel a indicating that in the presence of Zeb1 knockdown the expression of the mesendodermal genes was upregulated in ground state culture compared to control cells. As suggested by the Reviewer, the regulation of these genes has now been studied in Zeb1 knockdown cells and in all the experimental conditions tested.*

Q3. *Lastly, the manuscript contains numerous mistakes in the description of figures, numbering/labeling of panels, definition of culture conditions, etc. While these issues are usually considered "minor comments," they are not trivial in this work, particularly considering the importance of well-defined culture conditions in ES work.*

A3. *The authors thank the Reviewer for pointing out our mistakes. They have been fixed in the current manuscript version.*

Q4. *The study deals with an interesting topic for which there is a number of published articles. The work follows up of previous paper by the same authors (that surprisingly it is not cited) in Spallotta et al (2010) Stem Cells, 28:431-42. The manuscript also overlooks two articles directly related to the topic, namely: Mora-Castilla et al (2010). Cell Death & Diff, 17:1025-1033 and Tejedro et al (2010). Cell Death & Disease 1:e80. In their discussion of Mesp2 and Zeb1, authors should also cite and discuss J Cell Biol (2016) 4:463-7. There is also evidence in the literature that Zeb1 is not regulated by NO in epithelial but these works are not cited either.*

A4. *Literature has been updated according to Reviewer's suggestions. Spallotta et al (2010) Stem Cells, 28:431-42 is reference number 6; Mora-Castilla et al (2010). Cell Death & Diff, 17:1025-1033 is reference number 3; Tejedro et al (2010). Cell Death & Disease 1:e80 is reference number 14; and Chiapparro et al J Cell Biol (2016) 4:463-7 is reference number 36.*

Q5. *The definition of the media and culture conditions is confusing. In addition, in several experiments, some culture conditions are not included without explanation (see points 3 and 4). Obviously, this is a key issue when working with ES. For instance in line 66, authors indicated that they use FCS but in Methods they talk about FBS. In other example, two different culture conditions are labeled with the same name. In line 72-73: "in response to LIF withdrawal (DMSMNO), an effect further enhanced in the presence of NO (DMSMNO)". One has to assume that it is meant DMSM but this mistake only adds to the confusion.*

A5. *The culture conditions adopted in our work are relatively standard, however, in the revised version of the manuscript they have been provided in greater detail. Specifically, all the experiments have been performed in FBS while FCS has never been used: it has been just a typing error. For clarity a new definition of the experimental conditions adopted has been now introduced: SM (standard medium → DMEM+LIF+FBS 10%); GS (Ground state → NDiff N2 B27+GSK3i XVI+MEKi+LIF); DM (spontaneous differentiation medium → DMEM+ FBS 10%); NO (mesendodermal differentiation medium → DMEM+ FBS 10%+DETA/NO 250uM). A similar definition has been now added to the Methods section.*

Q6. *In Supplemental Figure 1, the authors show the differences between SM and GS cultures with regard to NO*

production. However, they do not measure NOs levels before NONOate administration as well as after 24hrs of NONOate administration.

A6. The authors would like to apologize for the misunderstanding. Regarding NO production, in fact, suppl. fig. 1 does not address NO production in SM and GS culture conditions. Data represent, instead, the assessment of cellular response to an exogenously added source of NO, as also stated in the related supplemental figure legend. The results shown in suppl. fig. 1 have been generated to provide basis for the ground state (GS) culture condition in all subsequent experiments.

Q7. In Supplemental Figure 2a, the levels of Zeb1 under SM culture conditions should be shown.

A7. Zeb1 level analysis under standard medium (SM) culture conditions has been performed and now shown in the updated suppl. fig. 2, panel a left. The effect of SM on mES was not further explored and all subsequent experiments were performed in GS, DM or NO conditions.

Q8. Authors claimed that in Supplemental Figure 2, panel b, DMGSNO "Zeb1 clearly relocate out the nucleus". Despite the authors claim that Zeb1 "clearly" relocates, I am not convinced by the data shown. At best, this figure only confirms the decrease in Zeb1 expression shown in "panel a" but the relocation of Zeb1 is not convincing. Most cells in the DMGSNO condition express little Zeb1 (if any at all). In sum, their data do not allow to conclude that Zeb1 is translocated. Such claim should be backed up by a western blot of nuclear and cytoplasmic fractions.

A8. Thanks to the Reviewer for the important suggestion. New experiments addressing Zeb1 relocation to the cytoplasm have now been added in the new suppl. fig. 2, panel c. The results show that Zeb1 is enriched in the cytoplasm of fractionated mES cells exposed to an exogenous source of NO.

Q9. Figure 3, panel a. It is not clear why the authors do not shown the DMGSNO condition at 1 h and 2h as they show these time points for the other culture conditions. The explanation of this Figure in the main text (line 163) indicates "gray stripped bars" but there are not gray stripped bars in Figure 3, panel a.

A9. In our work the effect of DETA/NO on the endogenous production of cGMP was evaluated in DM^{GSNO} (now renamed NO condition in the revised manuscript) merely as an internal control experiment. This point has been clarified in the revised manuscript version. We thank the Reviewer for raising the point about the gray stripped bars. However, this has been a printing error related to a previous version of the figure. The revised figure 3, panel a now shows the appropriate bar code symbols.

Q10. The efficiency of CRISP/Cas9 targeting of Zeb1 in Supplemental Fig 4c is very poor and questions the conclusions that authors make out of the targeting experiments. It would be essential to obtain better deletion of Zeb1 (see also below).

A10. We thank the Reviewer for the important comment. See answer to question 2 and suppl. fig. 4, panel a left.

Q11. Supplemental Fig 3b shows that Zeb1 was enriched in selected genes in the GS culture condition. Surprisingly, none of these genes is examined with the CRISP/Cas 9 targeted cells (LCv2_Zeb1_1 and LCv2_Zeb1_2). In addition, regulation of these genes by Zeb1 should be validated by quantitative PCR.

A11. Figure 4, panel a shows the results of new experiments performed following Reviewer's suggestion. As stated in the answer to question 2 (see above) we have now verified the effect of Zeb1 knockdown on the mesendodermal gene expression.

Q12. Very importantly, and key to justify their conclusions: the authors should analyze the differentiation genes in Figures 3 panel g and Figure 5 using the cells CRISP/Cas9 targeted for Zeb1 and Hdac2 and under the different culture conditions (GS, DMGS, DMGSNO).

A12. Thanks to Reviewer suggestion we verified the effect of Zeb1 or Hdac2 knockdown on the mesendodermal gene expression. The new results are now shown in the revised figure 4. However, we would like to emphasize here that the suggested experiments related to Figure 5 are not fully justified. In fact, the genes indicated by the Reviewer are late cardiovascular differentiation markers. They are not expressed in an early precursor population. The authors are not claiming, in fact, that Zeb1 or Hdac2 is regulating late genes neither made any specific statement in this direction.

Q13. Likewise, they should examine the expression of these genes in the cells targeted for Zeb1 and Hdac2 with overexpression of eNOS. Without those experiments, it is difficult to claim that Zeb1 is regulating these genes, let alone including such claim in the abstract and title of the manuscript.

A13. *The authors would like to apologize for the lack of clarity. However, the suggested experiments are unlikely to be justified. In fact, in conditions in which Zeb1 or Hdac2 is knocked-down eNOS appears expressed and well detectable already in ground state condition (see figure 3 panel h and i). Nevertheless, in the revised manuscript the authors performed new experiments in which wild type eNOS and its constitutively active eNOS^{S1177D} mutant have been overexpressed in mES cultured in GS condition (figure 5, panel c). In this condition, Zeb1 is well detectable as shown in suppl. Fig. 2. The results, now shown in the new figure 5 panel c, d and e, indicate that eNOS overexpression or that of its constitutively active mutant is associated with the presence of mesendodermal gene transcripts detectable in spite of GS suggesting for a positive role of eNOS in the regulation of those targets.*

Q14. *Authors should both characterize Zeb1 expression and effect of Zeb1 elimination in eSNO- and eSNO+ cells.*

A14. *A new experiment has been performed as shown in the revised panel e of figure 6. The result suggests that, as expected, Zeb1 expression is reduced in ESNO+ cells possibly as consequence of miR200b upregulation. Therefore we felt no strong rationale to perform an experiment aimed at eliminate Zeb1 from ESNO+ cells. Moreover, CRISPR/cas9 experiments aimed at targeting Zeb1, shown in the revised figure 3 panel g and figure 4 panel a, indicated that eNOS and the mesendodermal genes became already detectable in the presence of GS. Hence, we felt that there was no rationale exploring further the role of Zeb1 in ESNO- cells.*

Q15. *It should be demonstrated that the elimination of Zeb1 in eSNO- cells increases differentiation and miR-200 levels.*

A15. See A14.

Q16. *In addition, miR-200 should be knockdown in eSNO+ cells to inhibit differentiation*

A16. *In order to address the role of miR 200 family in mES cells, we knocked-down miR200b in wild type mES and demonstrated that Zeb1 expression remained elevated in differentiated cells. In the same experiment we observed a significant reduction of mesendodermal gene expression as expected in consequence of an elevated Zeb1 cellular content. The new data have been now shown respectively in the revised version of suppl. Fig. 2, panel d and suppl. Fig. 3, panel c.*

Q17. *Zeb1 should be overexpressed and examine the number of beating EBs formed.*

A17. *Zeb1 is expressed in ESNO- cells (figure 6, panel e) that do not generate beating EBs (figure 6 panel h). Therefore we felt no rationale to perform the suggested analysis. However, in order to understand further the effect of Zeb1 on mES differentiation we overexpressed the factor in wild type cells and found that they were unable to properly differentiate. Specifically, the forced expression of Zeb1 in naive mES reduced or prevented the expression of mesendodermal markers in cells released from stemness and exposed to NO. The results are shown in the new supplemental figure 5 of the revised manuscript version.*

Q18. *My understanding is that Zeb1 knock out mice does not have defects in cardiovascular development (Takagi et al., Development 125, 21-31,1998). Authors should at least discuss how their results fit with the phenotype of these mice. In light of the phenotype of the Zeb2 knock out mice, I feel it is unlikely that this could be explained either by compensation between Zeb1 and Zeb2.*

A18. *The authors would like to thank the Reviewer for the important suggestion. According to literature Zeb1 KO has an effect on neuroectodermal and neural-crest derived structures including some skeletal muscle components (Higashi et al. J Exp Med 1997; Takagi et al., Development 1998; Bellon et al., J Cell Biol 2009). Alterations in the cardiovascular system are poorly represented and limited to a sporadic hemorrhagic phenotype. The literature related to this point has been commented in greater detail in the Discussion section of the present version of the manuscript. This is in line with the evidence that Zeb1 physiologically counteracts the mesendodermal differentiation of mES in favour of neuroectodermal markers expression. Further, we found that ESNO- Zeb1 positive cells are enriched of neuroectodermal associated genes, such as Meis1 and Pax6 (see figure below).*

ESNO+ cells are not enriched for neuroectodermal marker expression. mRNA expression analysis of Meis1 and Pax6 in ESNO- (black bars) and ESNO+ (white bars) derived EBs taken at different time points of the EB maturation process. Data are shown as mean \pm SE fold increase compared to ESNO- derived EBs after subtraction of the housekeeping p0 gene signal (n=4 each time point; *p < 0.05).

Other comments

- 1) Several figures and panels are mislabeled. This is particularly the case of Figure 4. The panels explained and referred in the main text (lines 215-219) do not correspond to the panels shown in Figure 4.
- 2) While Supplemental Figures are numbered and labeled, the main Figures (at least the version that this Reviewer received) are not
- 3) There is no information about the identity of the Ab used in the Zeb1 ChIP-seq, beyond that is a Santa Cruz antibody. This is a highly relevant piece of information considering that the quality of the Ab used is essential in ChIP-seq analysis.
- 4) Line 253. The correct name of the GSK3 inhibitor is CHIR, not CHYR.
- 6) Mouse gene HDAC2 should be written as Hdac2.

The revised manuscript has been corrected according to Reviewer's suggestion

Reviewers' comments:

Reviewer #1 (Remarks to the Author):

The observations are interesting and the manuscript has been improved by the changes.

I have a couple of minor questions and comments

1. Is the effect of NO on PP2A (mentioned in the introduction) mediated by cGMP or S-nitrosylation (the latter has been reported). It's not clear as written.
2. Discussion first para, The authors refer to 'a dose effect of NO'. This is a pharmacological effect, not physiological. There is little evidence for dose effects of NO in any physiological context. "Dose effects" likely reflect engagement of alternative targets. I would remove this statement.

Reviewer #2 (Remarks to the Author):

The revised version of the manuscript has fixed, in some cases without a clear explanation of how they did it, some (and only some) of the minor issues. Nevertheless, my main concern regarding the lack of novel mechanistic insights with what has been already published still holds. The authors continue to overlook important earlier studies on the very same topic and to clarify the existing controversies between their findings and some of those studies. Lastly, several articles on the role of Zeb1 and Hdac1 (but, importantly, not Hdac2) in the differentiation of hESC/iPSC towards cardiomyocytes have been published since the original submission over a year ago. However, and once again, the authors failed again to cite any of these new works.

- 1) The study continues to overlook many works directly related to this study and to clarify existing controversies. A) PNAS 18:8242 showed that Hdac1, but not Hdac2, is involved in the differentiation of embryoid bodies into cardiomyocytes (and in the regulation of Nkx2-5 and Mef2c). That study found that HDAC1 is required for the reprogramming of ES cells upon differentiation and that conditional KO stem cells where Hdac1 and Hdac2 have been inactivated, only Hdac1-deficient embryoid bodies showed the spontaneous rhythmic contraction and increased expression of both cardiomyocyte and neuronal markers. J Biol Chem 280:19682 has also studied the role of HDACs in the differentiation of ES toward cardiomyocytes. These studies were not cited. The article in PNAS should be at least discussed as it directly relates to this study. B) Links between Hdac, eNOS and Zeb1 have been previously reported, not only in Gut 61:329 (cited in the manuscript), but also in EMBO Mol Med. 7:831, Stem Cells 31:1749 and many others. However, these articles were not cited or discussed. C) A recent study published in Circulation Research (doi:10.1161/CIRCRESAHA.116.310456) by Snyder's group has elegantly and convincingly shown the role of Zeb1 in cardiomyocyte differentiation. Using an RNA-seq of hiPSCs and hESCs during cardiomyocyte differentiation, the study in Circulation Research found that ZEB1 is required for early cardiomyocyte differentiation. The study goes well beyond this manuscript as it shows a much more complex picture of the transcriptomics involved at different stages

during cardiomyocyte differentiation, such as T and EOMES (mesoderm); LEF1 and MESP1 (from mesoderm to cardiac mesoderm); MEIS1 and GATA4 (post-cardiac mesoderm); JUN and FOS families, and MEIS2 (cardiomyocyte). The manuscript has a clear overlapping with this manuscript and authors should cite and discuss this important work. D) Lastly, while the authors now cited the four articles I highlighted in the first review, they have not engaged in any meaningful discussion on how these articles relate/overlap with their own work.

2) I continue to be concerned with the CRISP/Cas9 targeting of Zeb1. In the original manuscript (Supplemental Figure 4c) authors failed to obtain an efficient CRISP/Cas9 targeting of Zeb1 using two independent vectors (GS-LCv2_Zeb1_1 and GS-LCv2_Zeb1_2). Now (Supplementary Figure 4a) they show that CRIPS/Cas9 targeting has abrogated Zeb1 expression but it is not clear whether the CRIPS/Cas9 vector has been changed. The authors now referred this vector as LCv2_Zeb1. However, in Methods, the authors still mentioned the two vectors used in the original version (GS-LCv2_Zeb1_1 and GS-LCv2_Zeb1_2). One has to assume that only one of the original vectors worked. If that In that case, it is not clear which of the two vectors is shown in Supplementary Figure 4A and used in Figures 3g and 4a. It is not clear which targeting vectors referred in Methods are used in the manuscript, why only one of the two (or probably none of them) is shown in Suppl Fig 4c, which of the targeting vectors worked or why in Supp Fig 4c one of the targeting vectors has been now deleted. In any case, as standard practice (particularly for a journal like Nature Communications), at least TWO independent CRIPS/Cas9 targeting vectors for Zeb1 should be used, particularly considering that Zeb1 is the main point in the title and Abstract.

3) Cytoplasmic translocation of Zeb1. In the original version, the authors claimed that in Supplemental Figure 2, panel b, Zeb1 “clearly” relocate out the nucleus. A new Supp Fig 2c showing Zeb1 expression in cytoplasmic and nuclear fractions has been added. However, the authors made no effort to show a more representative immunofluorescence picture. Supp Fig 2b continues to be unconvincing at best. In other words, the results in Suppl Fig 2b and 2c do not match.

4) Figure 6e. The Western blot for Zeb1 in ESNO+ has half of the band stronger than the other half as if a bubble interfered with the blotting. Another blot should be shown.

5) For all the Western blots in the manuscript, full uncut gels should be shown. This a standard practice nowadays and it becomes critical in this manuscript where many of the blots shown were cut just below/above the band or the bands are not very clear (bottom of Fig 3e, Fig 5d, Fig 6a, Supp Fig 1b, Supp Fig 2d, Supp Fig 4a, Supp Fig 5a). Full uncut gels should be shown for all Western blots.

Minor comments

6) In my first review, I pointed to the need to provide a comprehensive list with the source of all the Abs used. This is critical considering that the quality of the Ab used is essential in ChIP-seq analysis. The authors indicated that the information has been now included. But

this reviewer was unable to find (unless I did not get all the manuscript documents) where this information has been included. At least, it is not in the section of Methods related to Western blot and immunoprecipitation.

Kind suggestion

7) Considering that: a) the data on Zeb1 are the weakest and, for the most part unconvincing, b) the article in *Circulation Research* overlaps with this manuscript and nicely defines a role for Zeb1 in cardiomyocyte differentiation, and c) the Hdac1/Hdac2 controversy (PNAS 18:8242), it is my humble opinion that the manuscript will benefit from packaging the data in a different manner.

Reviewer #3 (Remarks to the Author):

Cencioni et al. report that endothelial nitric oxide synthase (eNOS) is rapidly upregulated in a subpopulation of mouse ES cells upon exit from ground state pluripotency in 2i/LIF conditions. This eNOS-positive fraction produces nitric oxide (NO), shows elevated expression of mesendodermal genes and more efficiently generates embryoid bodies with beating areas. ES cells in which eNOS was disrupted by CRISPR-mediated gene targeting failed to produce NO or efficiently upregulate mesendodermal genes following the withdrawal of 2i/LIF. Mechanistically, NO was shown to inhibit the nuclear function of Hdac2 via S-nitrosylation, and to activate the expression of miR-200 miRNAs. Both of these events are proposed to antagonize the transcriptional repressor Zeb1, which normally suppresses mesendoderm-associated genes including eNOS under ground state conditions. The existence of a feedback circuitry among Zeb1, Hdac2 and eNOS is corroborated by additional genetic studies showing that CRISPR-mediated ablation of Zeb1 or Hdac2 in ES cells causes premature expression of eNOS and other mesendoderm-associated genes.

This manuscript uncovers a novel regulatory axis mediating mesendodermal commitment of ground state mouse ES cells. Previous work had suggested that low doses of NO delay differentiation (Tejedo et al., 2010), but this study demonstrates that eNOS/NO promote ES cell differentiation, especially into cardiovascular precursors. Overall the manuscript is of interest and appropriate for *Nature Communications*, but the authors should clarify the following points:

1. ChIP experiments indicate that Zeb1 becomes uncoupled from chromatin at mesendodermal target genes in differentiation media, regardless of whether an exogenous source of NO is supplied (Figure S3b). However, the induction of mesendodermal target genes was far more dramatic in the presence of exogenous NO (Figure S3a). This raises an important question which the authors do not comment on: how does exogenous NO further boost the expression of mesendodermal genes? Since Zeb1 is already fully uncoupled from chromatin in DM alone, there must be some additional mechanism whereby exogenous NO promotes mesendodermal commitment that is independent of Zeb1 (see also my next point).

2. The authors propose two distinct mechanisms for NO-dependent regulation of mesendodermal target genes: S-nitrosylation of Hdac2, which leads to nuclear export of Zeb1, and activation of miR-200b, which inhibits Zeb1 translation. While inhibition of miR-200b abrogates NO-dependent activation of several mesendoderm-associated genes, the activation of Wnt3 and Tbx4 was notably less affected (Figure S3c). Interestingly, these two mesendodermal targets were not induced in Zeb1-KO ES cells cultured with exogenous NO (Figure 4a), but they were highly induced in Hdac2-KO ES cells cultured in the same condition (Figure 4b). These observations suggest that Hdac2 mediates transcriptional repression at a subset of mesendodermal targets, such as Wnt3 and Tbx3, independently of Zeb1. Have the authors considered the possibility of such a Zeb1-independent role of Hdac2?

3. As a general point, I wonder how representative the experiments with an exogenous source of NO are for the endogenous activation of eNOS/NO that occurs during ES cell differentiation. Can the authors directly compare the levels of NO under these two experimental conditions? In addition, does the DAF+ subpopulation shown in Figure 6b have increased S-nitrosylation of Hdac2 (as was shown in Figure 3c in presence of exogenous NO)?

Minor points:

1. Throughout the manuscript CRISPR-targeted ES cell lines should be referred to as knockout, rather than knockdown clones. The term "knockdown" is typically used to denote RNAi studies where only partial downregulation of the target gene is achieved.
2. Please add in-figure labels to the ChIP-Seq tracks in Figure 2b and Figure S4b.

The Authors would like to thank the Editors for their interest in the present work.

REVIEWER 1:

Reviewers' comments:

The observations are interesting and the manuscript has been improved by the changes.

The authors would like to thank the Reviewer for His/Her positive comments to their revised work

Q1) Is the effect of NO on PP2A (mentioned in the introduction) mediated by cGMP or S-nitrosylation (the latter has been reported). It's not clear as written.

A1) The indicated point has been clarified in the Introduction as requested (see Page 5, lines 99-104).

Q2) Discussion first para, The authors refer to 'a dose effect of NO'. This is a pharmacological effect, not physiological. There is little evidence for dose effects of NO in any physiological context. "Dose effects" likely reflect engagement of alternative targets. I would remove this statement.

A2) The specific paragraph in the Discussion section has been amended according to Reviewer's suggestion (see Pages 14-15, lines 397-410).

REVIEWER 2:

Q1) PNAS 18:8242 showed that Hdac1, but not Hdac2, is involved in the differentiation of embryoid bodies into cardiomyocytes (and in the regulation of Nkx2-5 and Mef2c). That study found that HDAC1 is required for the reprogramming of ES cells upon differentiation and that conditional KO stem cells where Hdac1 and Hdac2 have been inactivated, only Hdac1-deficient embryoid bodies showed the spontaneous rhythmic contraction and increased expression of both cardiomyocyte and neuronal markers. J Biol Chem 280:19682 has also studied the role of HDACs in the differentiation of ES toward cardiomyocytes. These studies were not cited. The article in PNAS should be at least discussed as it directly relates to this study.

A1) The authors would like to thank the Reviewer for His/Her suggestion about updating the relevant literature concerning the role of Zeb1 and Hdacs in their system. However, we would like to raise here the point that as stated into the journal instructions to authors the number of allowed references is limited. Therefore only the most informative citations have been reported so far. For this reason, in the revised version of our work the indicated PNAS and other few references have been added and commented where appropriate. Hence, the suggested J Biol

Chem 280:19682 has not been cited in light of the fact that the functional studies described there were accomplished by means of the chemical pan-Hdac inhibitor Trichostatin A (TSA) whose effect has been matter of prior investigation in our laboratory (see ref. 6 in the revised manuscript) and is out of the scope of the present manuscript. Nevertheless, we took great advantage from the Referee' suggested citation of Cowley's work (PNAS 2010) reporting about a distinct role of Hdac1 and 2 in early mouse ES differentiation. The specific reference (now Ref number 21) has been added and commented in the Introduction and Discussion sections of the revised manuscript (Page 5 lines 108-120; Page 15 lines 433-448). The indicated article deals with a very relevant aspect related to the definition of Hdac1 and 2 functional specificities. The authors described that Hdac1 inactivation but not that of Hdac2 enhances cardiovascular differentiation in mouse ES. This is an interesting report that shed some light on the controversial matter of Hdac1/2 redundancy vs. specialized functions (e.g. Montgomery RL GENES & DEVELOPMENT 2007, 21:1790–1802; Hagelkruys A. Development 2014, 141: 604-61; Jamaladdin S. PNAS 2014 8, 111:9840-5). We believe of interest here to point out that Hdac1 and 2 share the same co-repressor complexes (see Figure 1).

[redacted]

Figure 1. A summary of the three main Hdac1/2-containing complexes: Sin3, NuRD and CoREST Each complex contains multiple subunits with DNA/chromatin-recognition motifs indicated by the color code. In addition, deacetylase and ELM2/SANT domains are also shown. The Hdac1/2-binding subunit within each complex, i.e. Sin3A/B, MTA1/2/3 or CoREST1/2/3, are shown in bold. (adapted from Biochem. Soc. Trans. (2013) 41, 741–749).

In consideration of the relatively ubiquitous distribution of these complexes and the complexity of their regulatory networks it is conceivable that at least some of Hdac1 and 2 functions could be overlapping as well as distinct effects will appear in specific contexts. Indeed the evidence provided by Cowley and coworkers does not rule out whether Hdac2 exerts any regulatory

functions on *Hdac1* (see also Hagelkruys A. et al. *Development* 2014;141, 604-616) or being required for its appropriate activity as also indicated by a more recent work from the same authors (see reference 22 in the revised manuscript version). In this regard, it is relevant to point out that specific post-translational modifications may occur differentially on *Hdac1* and *Hdac2* in spite of a large degree of sequence homology between the two molecules. Specifically, *Hdac1* is acetylated, phosphorylated, SUMOylated and poly-ubiquitinated while *Hdac2* is acetylated, phosphorylated, SUMOylated, carbonylated and *S*-nitrosylated (Figure 2). Notably, *Hdac2* and *Hdac6* are the only Hdacs known to be functionally controlled by nitric oxide through *S*-nitrosylation.

Figure 2.

Schematic representation of the PTMs spatial distribution of *Hdac1* and *Hdac2*. (a) Organization of the functional domains of *Hdac1* and *Hdac2*. Numbers indicate the corresponding amino acid (aa) positions. Percentages of identity were calculated by BLAST alignment of *Hdac1* (CAG46518.1) and *Hdac2* (AAH31055.2) protein sequences. (b) and (c) Visual comparison of the different PTMs occurring on *Hdac1* and *Hdac2* in the C-terminal (B) and central (C) domains. Different PTMs are illustrated by different colors and letters: P (yellow): phosphorylation, A (green): acetylation, U (blue): ubiquitination, S (red): SUMOylation, N (orange): nitrosylation, and C (lilac): carbonylation. The amino acid sites of modification are indicated by the number of their position in the protein sequence: S: serine, K: lysine, Y: tyrosine, C: cysteine, and CBP: CBP histone acetyltransferase. When the precise position of a modified residue is not known, the amino acid site is followed by an x. The question mark (?) is used when the precise amino acid site has not been formally identified. The amino acids at the

boundaries of the two domains are reported. (J. Biomed. and Biotechnol .doi:10.1155/2011/690848)

In agreement, our prior work (see references 16 and 19 in the revised manuscript reference list) demonstrated that Hdac2 but not Hdac1 is post-translationally modified by nitric oxide-dependent nitrosylation on cytosines 262 and 274 and that this modification led to Hdac2 detachment from DNA and functional inactivation. Noteworthy, ref.16 also reports that Hdac2 S-nitrosylation reduces Hdac1 activity (Figure 3, panels A, B, D, F).

[redacted]

Figure 3. NO regulates the enzymatic activity of Hdac2 by cysteine-S-nitrosylation. (A) Western blotting analysis of Hdac1, Hdac2, and Hdac3 in C2C12 myoblasts after immunoprecipitation with anti S-nitroso-cysteine-specific antibody. An increased cysteine-S nitrosylation of Hdac2 was detectable after 4h of treatment with DETA-NO. (B) The bar graph shows the Hdac1-, Hdac2-, and Hdac3-specific activity in C2C12 evaluated in the presence or absence of DETA-NO treatment for 4 h. (C) Anti-S-nitroso-cysteine antibody immunoprecipitation and Western blotting analysis of Hdac1, Hdac 2, and Hdac3 in WT and MDX mice infected with either eNOS S1177A or GFP adenovirus adductor muscles. An increased cysteine-S-nitrosylation of Hdac2 was detectable after 14 days from AdeNOS infection. (D) The bar graph shows in vivo the Hdac1-, Hdac2-, and Hdac3-associated enzymatic activity evaluated in adenovirus-infected WT and MDX mice expressing GFP or eNOS, respectively, in the adductor muscles. Assays were performed on total muscle lysates after immunoprecipitation with anti- Hdac1-, anti- Hdac2-, and anti- Hdac3-specific antibodies. (E) Evaluation of NO effect on recombinant *E. coli* GST-purified Hdac1-, Hdac2, and Hdac3-specific activities in the presence or absence of Hdac inhibitor SAHA (SA, 5 μ M) or NO donors SNAP (SN, 205 μ M) or GSNO (GS, 250 μ M). (F) Western blotting analysis of S-nitroso-cysteine on recombinant Hdac1 and Hdac2 proteins after NO donors SNAP (SN, 10 mM) or GSNO (GS, 10 mM). Data were normalized by densitometry analysis (Right). (PNAS 2008, 105:19183-7).

In light of this evidence we cannot exclude that Hdac1 could be negatively influenced by the S-nitrosylation of Hdac2. To further explore this point, in the revised manuscript version, new experiments have been provided showing that in experimental conditions in which Zeb1 and Hdac2 are detached from chromatin also Hdac1 detaches from the promoter regions of Zeb1 target genes involved in the regulation of mesendoderm/cardiovascular commitment. The results are now shown in suppl. Figure 3 panel c and d suggesting for the presence of a coregulatory mechanism among Zeb1, Hdac1 and Hdac2.

A specific comment addressing this aspect has been included to the revised version of our manuscript in which a sentence about a possible indirect and negative effect of NO on Hdac1 has been added in the Results and Discussion sections (page 10, lines 260-265; page 15 lines 432-438).

Q2) Links between Hdac, eNOS and Zeb1 have been previously reported, not only in Gut 61:329 (cited in the manuscript), but also in EMBO Mol Med. 7:831, Stem Cells 31:1749 and many others. However, these articles were not cited or discussed.

A2) The authors are aware that Zeb1, Hdac1 and 2 interact forming a regulatory complex as depicted in Figure 4 functionally relevant in the process of epithelial/mesenchymal transition.

[redacted]

Figure 4. Zeb1 is an important repressor of the CDH1 gene in pancreatic cancer. In epithelial pancreatic cancer cells, transcription of the CDH1 gene, encoding E-cadherin, is activated by a multiprotein complex containing transcription factors (TF) which recruit RNA polymerase II (Pol II) via a coactivator complex (CAC) to the CDH1 promoter. During the dynamic process of epithelial to mesenchymal transition (EMT), the CDH1 gene is epigenetically silenced. One pathway involves the transcriptional repressor Zeb1, which recruits Hdac1 and/or Hdac2 containing corepressor complexes (CRC) to inhibit transcription of the CDH1 gene. **Gut 2012;61:329e330.**

However, the interconnection among Zeb1, Hdacs and NOS isoforms is still controversial and largely uncharacterized. In a manuscript from Rossig et al (Circ Res. 2002;91:837-44) it is reported that eNOS regulation by Hdacs may occur at transcription and post-transcription level while no clear association has been made between Zeb1 and eNOS expression (see additional comments below). Regrettably, space limits our possibility to report all the redundant citations related to this matter and in our revised manuscript we were forced to make a choice in favor of those strictly relevant for our findings. The work of Adhgassi et al on Gut 61:329, is, to our knowledge, one of the earliest contribution describing the protein-protein association of Zeb1 with both Hdac1 and 2. Unfortunately, we could not find a way to cite the work from Meidhof S. et al (EMBO Mol.Med. 2015;7:831-47) in our manuscript. This is in consequence of the fact that in that study chemical inhibitors of Hdac activity were used to investigate Zeb1 biological properties. Their characterization is out of the scope of our present study. In fact, we addressed the role of Hdac inhibitors in one of our prior publications on mES (see reference 6 in the revised manuscript and figure 5 below).

[redacted]

Figure 5. HDACi differentially regulate ESC differentiation—immunofluorescence analysis. ESCs were deprived of LIF for 24 hours in the presence or absence of HDACi or control solvent (DMSO) and double immunofluorescence was performed. (A): This picture shows that TSA and MC1568 (specific class II Hdac inhibitor) prevent the spontaneous onset of Flk-1 and Desmin while inducing the expression of Nestin. On the contrary, MS27-275 (specific Class I Hdac inhibitor) induced the expression of Flk-1 and Desmin. This result is representative of five independent experiments. (B): This picture shows that TSA and MC1568 prevent the spontaneous onset of SM22a and Bry. On the contrary, MS27-275 induced the expression of these markers. This result is representative of five independent experiments. (see Spallotta et al ref. 6)

Specifically, in our prior work (see ref. 6) we reported that class I selective Hdac inhibitors were able to activate a mesendodermal differentiation program in mouse ES cells. In the revised version of the present manuscript, we updated our reference list including the manuscript from Luo et al Stem Cells 31:1749 which, interestingly, demonstrated the high level of expression of Zeb1 in undifferentiated hESC and the importance of Zeb1 as repressor of vascular precursor commitment in that system. This work is conceptually very close to ours and we reasoned that it would really enforce our findings (see also A3 below). In spite of this positive contribution, it is worth noting that in Luo's study the expression of eNOS is merely used as a vascular differentiation marker and no mechanistic evidence or associations with Zeb1 or Hdacs has been provided. On the opposite, in our manuscript (Figure 3, panel g) data in favor of a direct association between Zeb1 and eNOS promoter are provided contributing original information about the presence of a possible eNOS-Zeb1 regulatory loop in our system.

Q3) A recent study published in Circulation Research (doi:10.1161/CIRCRESAHA.116.310456) by Snyder's group has elegantly and convincingly shown the role of Zeb1 in cardiomyocyte differentiation. Using an RNA-seq of hiPSCs and hESCs during cardiomyocyte differentiation, the study in Circulation Research found that ZEB1 is required for early cardiomyocyte differentiation. The study goes well beyond this manuscript as it shows a much more complex picture of the transcriptomics involved at different stages during cardiomyocyte differentiation, such as T and EOMES (mesoderm); LEF1 and MESP1 (from mesoderm to cardiac mesoderm); MEIS1 and GATA4 (post-cardiac mesoderm); JUN and FOS families, and MEIS2 (cardiomyocyte). The manuscript has a clear overlapping with this manuscript and authors should cite and discuss this important work.

A3) The authors would like to thank the Reviewer for pointing out this important recent work published while our manuscript was under the second round of revision. The manuscript from Liu Q et al (Circ Res. 2017;121:376-391) has now been added to the Bibliography section of our revised manuscript as ref number 54. Nevertheless, after careful evaluation the authors concluded that, although cleverly performed, the work from Liu et al. had modest impact on their findings. In fact, i) Liu's work has been made in very different cellular systems not directly comparable to ours, e.g. human embryonic and human inducible pluripotent stem cells; ii) The results from Liu et al. are in apparent contrast with reports previously published by others (see below); iii) Consistently, the results of Liu et al are also in contrast with our functional data showing that forced expression of Zeb1 prevents/reduces expression of mesendodermal/cardiovascular markers as shown in supplemental figure 5. Considering the diversity in the experimental settings and the other evidences provided in literature, which are instead in agreement with our results, (see point ii listed above and the additional comments provided below) no clear explanation is possible at this time about this apparent discrepancy; iv) In our work, we aimed first at investigating molecularly the effect of pharmacological doses of nitric oxide on mES ending up with the identification and characterization of an unprecedented precursor cell population autonomously synthesizing NO; v) Liu and coworkers did not address this specific point, as they did not perform any experiment to investigate the role of NO during differentiation of human iPS and hESC.

More in detail, not only Liu’s work is in contrast with our present work but also in contrast with data reported in the work of Luo et al (Stem Cells 31:1749) and that of Kim Y et al (Int J Stem Cells. 2017;10:28-37). These other authors, in fact, stated respectively that:

- “we have provided strong evidence to support its transcriptional repressor’s role of Zeb1/TCF8 in EC gene regulation during EC differentiation from hES cells. Data from loss-of-function gene studies using specific Zeb1 siRNA pool showed that the knockdown of endogenous Zeb1 gene expression **significantly increased EC-specific gene expression levels and promoter activity.**” Luo et al. Stem Cells 31:1749.

And

- “**Downregulation of Zeb1, a direct target of miR-200 family, and E-CADHERIN, a target protein of Zeb1, was observed in hESCs during differentiation into endodermal and mesodermal lineages, respectively.**” Kim Y et al Int J Stem Cells. 2017;10:28-37.

Remarkably, our experimental findings are well supported by the two studies mentioned above to whom we added additional novel information. We have not clear explanation for the controversy or how to reconcile discrepancies among the work of different authors and laboratories. Further work is required to solve this matter. It is evident, however, that the experimental cellular models adopted in our and Liu’s work are very dissimilar (human vs murine ES) and that the pharmacological effect of nitric oxide has not been addressed in the recent Circulation Research manuscript making difficult any further comparison. However Liu’s results were commented in the Discussion section (Page 14; lines 390-397).

Q4) Lastly, while the authors now cited the four articles I highlighted in the first review, they have not engaged in any meaningful discussion on how these articles relate/overlap with their own work.

A4) As requested new comments have been provided related to the literature in object (ref 6 in the Introduction; ref 3 in the Introduction; ref 14 page 14 line 399-403; ref 41 page 13-14 lines 383-390).

Q5) I continue to be concerned with the CRISP/Cas9 targeting of Zeb1. In the original manuscript (Supplemental Figure 4c) authors failed to obtain an efficient CRISP/Cas9 targeting of Zeb1 using two independent vectors (GS-LCv2_Zeb1_1 and GS-LCv2_Zeb1_2). Now (Supplementary Figure 4a) they show that CRIPS/Cas9 targeting has abrogated Zeb1 expression but it is not clear whether the CRIPS/Cas9 vector has been changed. The authors now referred this vector as LCv2_Zeb1. However, in Methods, the authors still mentioned the two vectors used in the original version (GS-LCv2_Zeb1_1 and GS-LCv2_Zeb1_2). One has to assume that only one of the original vectors worked. If that In that case, it is not clear which of the two vectors is shown in Supplementary Figure 4A and used in Figures 3g and 4a. It is not clear which targeting vectors referred in Methods are used in the manuscript, why only one of the two (or probably none of them) is shown in Suppl Fig 4c, which of the targeting vectors worked or why in Suppl. Fig 4c one of the

targeting vectors has been now deleted. In any case, as standard practice (particularly for a journal like Nature Communications), at least TWO independent CRISPs/Cas9 targeting vectors for Zeb1 should be used, particularly considering that Zeb1 is the main point in the title and Abstract.

A5) *All experiments were performed with clones derived from both original vectors generating two independent CRISPR/Cas9 Zeb1 gene knockout. In the new version of the manuscript_R2, the authors added western blotting depicting the results obtained from both vectors (Supplemental figure 4 panel c). The related original nomenclature was restored: LCv2_Zeb1_1; LCv2_Zeb1_2; LCv2_Hdac2_1 and LCv2_Hdac2_2. New panels in the supplemental figure 4 panel c have been added to the supplemental set of revised figures and method section was explicated accordingly (Page 26 lines 774-780). The authors apologize for the lack of clarity in this matter.*

Q6) Cytoplasmic translocation of Zeb1. In the original version, the authors claimed that in Supplemental Figure 2, panel b, Zeb1 “clearly” relocate out the nucleus. A new Supp Fig 2c showing Zeb1 expression in cytoplasmic and nuclear fractions has been added. However, the authors made no effort to show a more representative immunofluorescence picture. Supp Fig 2b continues to be unconvincing at best. In other words, the results in Suppl Fig 2b and 2c do not match.

A6) *According to Reviewer' suggestion a new confocal analysis has been performed and a new picture has been added to the manuscript (see new supplemental figure 2, panel b).*

Q7) Figure 6e. The Western blot for Zeb1 in ESNO+ has half of the band stronger than the other half as if a bubble interfered with the blotting. Another blot should be shown.

A7) *A new and better quality western blot has been provided. See figure 6, panel e.*

Q8) For all the Western blots in the manuscript, full uncut gels should be shown. This a standard practice nowadays and it becomes critical in this manuscript where many of the blots shown were cut just below/above the band or the bands are not very clear (bottom of Fig 3e, Fig 5d, Fig 6a, Supp Fig 1b, Supp Fig 2d, Supp Fig 4a, Supp Fig 5a). Full uncut gels should be shown for all Western blots.

A8) *Uncut gels for all the westerns have now been provided as a supplemental set of figures (See file named: C Cencioni et al Mechanics of mesendodermal commitment in ground state mES_Supplemental full-lenght images of representative WBs_R2).*

Q9) In my first review, I pointed to the need to provide a comprehensive list with the source of all the Abs used. This is critical considering that the quality of the Ab used is essential in ChIP-seq analysis. The authors indicated that the information has been now included. But this reviewer was unable to find (unless I did not get all the manuscript documents) where this information has been included. At least, it is not in the section of Methods related to Western blot and immunoprecipitation.

A9) *An updated table with all information about the antibodies used has been added to the manuscript (see Table 4).*

REVIEWER 3

Reviewer #3 (Remarks to the Author): Cencioni et al. report that endothelial nitric oxide synthase (eNOS) is rapidly upregulated in a subpopulation of mouse ES cells upon exit from ground state pluripotency in 2i/LIF conditions. This eNOS-positive fraction produces nitric oxide (NO), shows elevated expression of mesendodermal genes and more efficiently generates embryoid bodies with beating areas. ES cells in which eNOS was disrupted by CRISPR-mediated gene targeting failed to produce NO or efficiently upregulate mesendodermal genes following the withdrawal of 2i/LIF. Mechanistically, NO was shown to inhibit the nuclear function of Hdac2 via S-nitrosylation, and to activate the expression of miR-200 miRNAs. Both of these events are proposed to antagonize the transcriptional repressor Zeb1, which normally suppresses mesendoderm-associated genes including eNOS under ground-state conditions. The existence of a feedback circuitry among Zeb1, Hdac2 and eNOS is corroborated by additional genetic studies showing that CRISPR-mediated ablation of Zeb1 or Hdac2 in ES cells causes premature expression of eNOS and other mesendoderm-associated genes. This manuscript uncovers a novel regulatory axis mediating mesendodermal commitment of ground state mouse ES cells. Previous work had suggested that low doses of NO delay differentiation (Tejedo et al., 2010), but this study demonstrates that eNOS/NO promote ES cell differentiation, especially into cardiovascular precursors. Overall the manuscript is of interest and appropriate for Nature Communications, but the authors should clarify the following points:

The authors would like to thank the Reviewer for His/Her positive evaluation of their work. The following is a point to point letter addressing the specific concerns.

Q1) ChIP experiments indicate that Zeb1 becomes uncoupled from chromatin at mesendodermal target genes in differentiation media, regardless of whether an exogenous source of NO is supplied (Figure S3b). However, the induction of mesendodermal target genes was far more dramatic in the presence of exogenous NO (Figure S3a). This raises an important question which the authors do not comment on: how does exogenous NO further boost the expression of mesendodermal genes? Since Zeb1 is already fully uncoupled from chromatin in DM alone, there must be some additional mechanism whereby exogenous NO promotes mesendodermal commitment that is independent of Zeb1 (see also my next point).

A1) *The effect of nitric oxide is epigenetically pleiotropic. In fact, our prior work (Illi et al. Pharmacol Ther. 2009;123:344-52) summarizes the properties of NO in terms of class I Hdac regulation as well as activator of protein phosphatases 2A (PP2A) (see also Illi et al. Circ Res. 2008;102:51-8; ref 17). Indeed, PP2A is involved in the regulation of class II HDACs whose activation and nuclear translocation is important for the mesendodermal commitment of mES as*

reported previously in our work from Spallotta et al Stem Cells. 2010;28:431-42 (ref 6). Moreover, NO is able to induce miR-200 family, an additional mechanism whereby exogenous NO contributes to mesendodermal differentiation of mES (Rosati et al ATVB 2011;31(4):898-907; ref 5). Mora-Castilla et al demonstrated that NO represses stemness marker genes such as Nanog and Oct4 as consequence of an increase in H3 trimethylation and an enhancement of pSer315 p53 (Cell Death Diff 2010;17:1025-33; ref 3). Furthermore, NO dependent increase of cGMP is able to increase the expression of cardiovascular markers like Nkx2.5 and myosin light chain (PNAS 2008;105:18924-29; ref 4). It is therefore well conceivable that additional mechanisms will take place during the NO-dependent regulation of gene expression occurring in differentiating mES. A new comment about this important aspect has been added in the Discussion of the revised manuscript version (pages 14, lines 397-422).

Q2) The authors propose two distinct mechanisms for NO-dependent regulation of mesendodermal target genes: S-nitrosylation of Hdac2, which leads to nuclear export of Zeb1, and activation of miR-200b, which inhibits Zeb1 translation. While inhibition of miR-200b abrogates NO-dependent activation of several mesendoderm-associated genes, the activation of Wnt3 and Tbx4 was notably less affected (Figure S3c). Interestingly, these two mesendodermal targets were not induced in Zeb1-KO ES cells cultured with exogenous NO (Figure 4a), but they were highly induced in Hdac2-KO ES cells cultured in the same condition (Figure 4b). These observations suggest that Hdac2 mediates transcriptional repression at a subset of mesendodermal targets, such as Wnt3 and Tbx3, independently of Zeb1. Have the authors considered the possibility of such a Zeb1-independent role of Hdac2?

A2) Following Reviewer' suggestion we now explored the enrichment of Hdac1, Hdac2 on the promoter of our gene signature and found that Zeb1, Hdac1 and Hdac2 similarly detached from target chromatin regions when cells were released from stemness or exposed to the NO donor. This evidence does not allow to assign Hdac2 alone a role in the regulation of Wnt3 or Tbx4 clearly distinct from that of Zeb1. Hence, the apparent positive consequence of Hdac2 inactivation on the expression of those two target genes must imply additional mechanisms possibly associated to the structure of their promoter and the recruitment of gene-specific transcription factors. The new data (Supplemental figure 3, panel d) and a comment have been added to the Results section of the revised manuscript (page 10 lines 261-266).

Q3) As a general point, I wonder how representative the experiments with an exogenous source of NO are for the endogenous activation of eNOS/NO that occurs during ES cell differentiation. Can the authors directly compare the levels of NO under these two experimental conditions?

A3) A new experiment has been performed addressing this specific issue. Supplemental figure 4, left panel a shows that in the presence of the NO donor DETANO a higher number of mES became DAF positive compared to cells released from stemness in the absence of an exogenous source of NO. The experiment however shows that during time-course the dynamics of NO signal was nearly similar in both conditions reaching a peak at 2 hours and declining with a similar fashion in the first 24 hours from induction. Interestingly, evaluation of the mean fluorescence

intensity (MFI) revealed no significant differences between the two populations in spite of the presence of the NO donor. This result, shown in figure 4 right panel a indicated that, in our experimental conditions, the global intracellular content of NO was similar in cells released from stemness in the presence or absence of NO. In summary, the NO donor increases the total number of DAF-positive cells however does not significantly affect the fluorescence intensity derived from the NO-dependent activation of DAF reporter probe. A comment about these new data has been added to the Result section of the revised manuscript version (page 10 lines 275-279).

Q4) In addition, does the DAF+ subpopulation shown in Figure 6b have increased S-nitrosylation of Hdac2 (as was shown in Figure 3c in presence of exogenous NO)?

A4) *A new experiment has been performed addressing this point. Figure 6 panel f shows that sorted ESNO+ cells present a significantly higher content of nitrosylated Hdac2 compared to ESNO- cells. This evidence reinforces the role of Hdac2 nitrosylation in the commitment of ESNO+ cells to mesendodermal differentiation. This result has been added to the revised Results section (page 13 lines 360-361) and commented into Discussion (page 17 lines 508-511).*

Minor points:

- 1. Throughout the manuscript CRISPR-targeted ES cell lines should be referred to as knockout, rather than knockdown clones. The term “knockdown” is typically used to denote RNAi studies where only partial downregulation of the target gene is achieved.**

Manuscript has been amended as indicated.

- 2. Please add in-figure labels to the ChIP-Seq tracks in Figure 2b and Figure S4b.**

Labels have been added as requested.

Reviewer #3 (Remarks to the Author):

The authors have provided detailed answers to the reviewers' questions and have performed additional experiments. However, I still have a number of concerns:

1. The authors have added Hdac1/2 ChIP experiments to Supplemental Figure 3c-d. These experiments show that, as noted for Zeb1 (Supplemental Figure 3b), Hdac1/2 becomes decoupled from chromatin at mesendoderm target genes upon switch to differentiation medium (DM), regardless of the presence of exogenous NO. However, with the exception of Dll1, most upregulation of mesendoderm target genes only occurred when DM was supplemented with exogenous NO (Supplemental Figure 3a). Thus, some mechanism other than Zeb1 or Hdac1/2 decoupling from these targets is responsible for stimulating these mesendodermal genes in the presence of NO. Another possibility is that Zeb1 or Hdac1/2 indirectly regulate these target genes by repressing an as-yet-unidentified transcriptional activator. As currently presented, this dataset is problematic because it does not support the authors' central argument that NO directly relieves the repression of mesendodermal genes by evicting Zeb1 and Hdac1/2 from chromatin at these particular loci. The authors do not directly comment on this issue in their revised manuscript. It may be possible to identify target genes that specifically lose Zeb1 or Hdac1/2 binding upon exposure to NO by performing ChIP-Seq for Zeb1 or Hdac1/2 in DM \pm NO.

2. Following on the previous point, the authors should be careful throughout their study to distinguish between those effects mediated specifically by exposure to NO vs. effects that arise simply from the switch to DM. For example, Fig. 1c highlights differentially expressed genes (DEGs) between mESCs maintained in 2i/LIF (GS) vs. mESCs exposed to DM with exogenous NO. However, to determine the response to NO, it would be more appropriate to focus on DEGs in DM \pm NO. Similarly, the "NO up-genes" shown in Fig. 2a currently also include genes that are upregulated in DM alone, i.e. without exogenous NO.

3. In several panels the authors compare mRNA expression levels of five mesendodermal genes in mESCs in GS vs. 24h after exposure to DM or NO. However, the mRNA fold increases in Supplementary Figure 3a are much greater than those shown in Supplementary Figure 3e or Figure 4a-b. For example, Tbx4 was induced ~500-fold after 24h of NO treatment in Supplementary Figure 3a, but Tbx4 upregulation was only ~20-fold in Scramble-LNA control cells (Supplementary Figure 3e) or LCv2_NTC control cells (Figure 4a) at the same timepoint relative to GS. How do the authors explain these differences?

4. Some paragraphs are currently too long and contain too many details, which makes the paper inaccessible to a general audience. I recommend introducing line breaks at the following positions to split up the long paragraph in the Results section on p.9: lines 229-230 (start a new paragraph with "To investigate whether Zeb1 could be involved...") and lines 253-254 (start a new paragraph with "The expression of the most represented NO-Zeb1-dependent genes...").

5. The manuscript must be properly edited to improve English grammar and spelling. The Discussion alone contained the following obvious typos: "heterogeinicity" (line 396),

"hypotesize" (line 435), "trascrition factor" (line 439), "repression transcription factor" (line 439, I suggest "repressive"), "metastatization" (line 464). The authors should also avoid terms like "stemness" (abstract, line 58) or "stemmed out" (Discussion, line 382), which are not very specific and instead refer to "self-renewal" or "pluripotency", where appropriate. The abbreviation "mES" is used throughout the text, but "ESC" or "ES cells" are more commonly used abbreviations.

The authors would like to thank the Editor and the Reviewer for their positive comments to our work.

Reviewer #3:

Q1. The authors have added Hdac1/2 ChIP experiments to Supplemental Figure 3c-d. These experiments show that, as noted for Zeb1 (Supplemental Figure 3b), Hdac1/2 becomes decoupled from chromatin at mesendoderm target genes upon switch to differentiation medium (DM), regardless of the presence of exogenous NO.

A1. We would like to thank the Reviewer for His/Her valuable comment. We understand that our data may not be easily interpreted as a consequence of lack of representation clarity. In our experiments mouse ESC begin synthesizing nitric oxide as soon as two hours after changing culture condition from self-renewal to differentiation (see Fig. 3a and Fig. 6b and suppl. Fig. 4a below and in the manuscript body). Hence, it may be unfeasible to fully separate the biological effect of DM “per se” from the culture medium supplemented with exogenous NO. In this condition, the evaluation of spontaneous production of NO has been one of the goals of our work that revealed, for the first time, the presence of a new mouse ESC sub-population spontaneously synthesizing NO upon release from pluripotency (See Fig. 6b below and in the manuscript body). Noteworthy, as requested by this Reviewer, in the revised manuscript version we were able to assess that the intensity of NO production from the endogenous cell population was similar to that obtained by addition of the exogenous source of NO (See suppl. Fig. 4a below and in the body of the manuscript).

Fig 3a. The graph shows the results of cGMP intracellular level quantification in mES cultured 24 h in GS (black bar), 30 min, 1h, 2h or 3h in DM alone or in the presence of PTIO, ODQ, L-NAME (gray bars) and 24h in NO alone or in the presence of PTIO (white bars), as indicated. Represented data are the mean of three independent experiments \pm s.e.m. (* $p < 0.05$ DM vs GS; ° $p < 0.05$ treatments vs DM).

Fig 6b. Left cytogram: representative scatter plot showing FSC-A and SSC-A distribution of the mESC population cultured in DM for 2 h in the presence of the DAF fluorescent probe. Right cytogram: representative dot plot showing the FITC and SSC-A distribution of the mESC population cultured in DM for 2 h in the presence of the DAF fluorescent probe. In each

experiment, gating to separate ESNO+ (DAF+; blue dots) from ESNO- (DAF-; red dots) cells was established manually (n=3).

Suppl fig 4a. Left panel: percentage of DAF positive mESC cultured 1, 2, 3, 6 and 24 h in DM or NO revealed by FACS analysis. Data are represented as mean±s.e.m. (n=3; *p<0.05 vs DM). Right panel: DAF mean fluorescence intensity (MFI) measured by FACS analysis in mESC cultured 1, 2, 3, 6 and 24 h in DM or NO. Data are represented as mean±s.d. (n=3).

Q2. However, with the exception of *Dll1*, most upregulation of mesendoderm target genes only occurred when DM was supplemented with exogenous NO (Supplemental Figure 3a). Thus, some mechanism other than *Zeb1* or *Hdac1/2* decoupling from these targets is responsible for stimulating these mesendodermal genes in the presence of NO.

A2. The authors apologize for the lack of clarity regarding their data representation in suppl. Fig. 3a. In the revised version of suppl. Fig. 3a (see below), the y axes has been now divided into two segments, making more easily noticeable that not only *Dll1* gene increases in DM condition but also other members of the signature, including *Wnt3*, *Wnt7b*, *Tbx4* and *Mesp2*. Furthermore, the NO synthesis inhibitor L-NAME added in DM reduced mesendodermal gene expression as compared to DM alone (see suppl. Figure 4d below and in the manuscript version). This observation is paralleled by the experimental evidence provided in Fig. 3b (see below and in the manuscript body) in which the NO scavenger PTIO fully reversed *Zeb1* binding to signature gene chromatin. In line with the above evidence, an independent series of experiments performed by using a constitutively active eNOS mutant (*eNOS*^{S1177D}) showed that the signature genes became already detectable in ground state condition further indicating that NO synthesis controls their expression (see Fig. 5e below and in the manuscript body). In the same condition, *Hdac2* was found nitrosylated (See Fig.5d below and in the manuscript body).

To further clarify the point raised by Reviewer a new set of experiments has been performed in which we were also able to evaluate that, in the presence of PTIO, the expression of our signature genes was significantly prevented in spite of the release of mESC from pluripotency (see fig. below that can be provided in a revised version of the manuscript as new suppl. Fig. 3b).

New Suppl fig 3a. mRNA expression analysis of Wnt3, Wnt7b, Tbx4, Dll1 and Mesp2 in mES cultured 24 h in GS (black bars), DM (grey bars) or NO (white bars). Data are represented as mean fold increase \pm s.e.m. compared to GS after subtraction of the housekeeping gene p0 signal (n=5; *p < 0.05 vs. GS; °p < 0.05 vs. DM).

Suppl fig 4d. mRNA expression analysis of eNOS, Wnt3, Dll1 and Wnt7b in mESC cultured 24 h in GS (black bars), DM (grey bars), DM + L-NAME (gray striped bars), DM + ODQ (gray squared bars) and NO (white bars). Data are represented as mean fold increase \pm s.e.m. compared to GS after subtraction of the housekeeping gene p0 signal (n=4; *p < 0.05 vs. GS; #p < 0.05 vs. DM).

Figure 3

Fig 3b. ChIP/qRT-PCR analysis of Zeb1 chromatin binding on nitric oxide-dependent mesendodermal genes: Wnt3, Wnt7b, Tbx4, Dll1 and Mesp2. Chromatins were extracted from mESC culture 24h in DM alone (grey bars) or in the presence of PTIO (striped bars). Data are shown as mean fold increase \pm s.e.m. compared to IgG value (striped bars) after Input normalization (n=3).

Fig 5e. qRT-PCR analysis of mRNA expression relative to the Wnt3, Wnt7b, Tbx4, Dll1 and Mesp2 transcripts in mESC cultured for 24h in GS prior (control Adeno virus Ad_GFP; black bar) and after infection by Ad_eNOSwt (gray bars) or Ad_eNOS^{S1177D} expressing the constitutive active eNOS (white bars). Data are shown as the mean of three independent experiments \pm s.e.m. represented as fold increase compared to Ad-GFP infected mESC after subtraction of the housekeeping gene p0 signal (* $p < 0.05$ vs Ad_GFP).

Fig 5d. Representative immunoprecipitation (IP) western blotting (WB) analysis of Hdac2 S-nitrosylation performed in mES infected by Ad_GFP, Ad_eNOSwt or Ad_eNOS^{S1177D} cultured in GS or after 2h from release into DM (n=3 each group).

New Suppl Fig 3b. mRNA expression analysis of Wnt3, Wnt7b, Tbx4, Dll1 and Mesp2 in mESC cultured 24 h in GS (black bars), DM (grey bars), DM + PTIO (gray striped bars). Data are represented as mean fold increase \pm s.e.m. compared to GS after subtraction of the housekeeping gene p0 signal (n=4).

Q3. Another possibility is that Zeb1 or Hdac1/2 indirectly regulates these target genes by repressing an as-yet-unidentified transcriptional activator. As currently presented, this dataset is

problematic because it does not support the authors' central argument that NO directly relieves the repression of mesendodermal genes by evicting Zeb1 and Hdac1/2 from chromatin at these particular loci. The authors do not directly comment on this issue in their revised manuscript.

A3. We thank the Reviewer for His/Her suggestion. We will add a paragraph in the Discussion section exploiting the possibility that an additional indirect mechanism could be involved in the regulation of mesendodermal genes. However, we would like to stress here that our central argument about the NO-dependent eviction of Zeb1 and Hdac1/2 from chromatin is still standing. In fact, this mechanism seems to play an important role in the activation of the mesendodermal transcription program described in our manuscript. Specifically, the following experimental evidences substantiate our conclusion:

1. Treatment with PTIO in DM condition rescued Zeb1 binding to chromatin at the mesendodermal loci (see again Fig. 3b below).
2. Hdac2 appears significantly nitrosylated as soon as mouse ESC are released from pluripotency (see Fig. 3c below and in the body of the manuscript) or in the presence of a constitutively active eNOS mutant (see Fig. 5d below and in the body of the manuscript) as apparent direct consequence of the endogenous NO production. As previously reported this modification prevents Hdac2 from chromatin binding (see references 18 and 19 in the manuscript: 18. Nott, A. et al. S-nitrosylation of HDAC2 regulates the expression of the chromatin-remodeling factor Brm during radial neuron migration. *Proc Natl Acad Sci U S A* 110, 3113-3118 (2013); 19. Nott, A., Watson, P.M., Robinson, J.D., Crepaldi, L. & Riccio, A. S-Nitrosylation of histone deacetylase 2 induces chromatin remodelling in neurons. *Nature* 455, 411-415 (2008)).
3. The knockout of Zeb1 by CRISPR/Cas9 technology promoted the expression of the signature genes already in cells cultured in ground state thus bypassing the repression effect of the 2i/L condition (See Fig. 4a below and in the manuscript body).
4. On the contrary, Zeb1 overexpression counteracted the effect of NO on the expression of the signature genes (see suppl. Fig 5a and b below and in the manuscript body).
5. The genetic inactivation of eNOS prevented the expression of the signature genes after mESC release from pluripotency (see Fig. 5b below and in the manuscript body).

Figure 3

Fig 3b. ChIP/qRT-PCR analysis of Zeb1 chromatin binding on nitric oxide-dependent mesendodermal genes: Wnt3, Wnt7b, Tbx4, Dll1 and Mesp2. Chromatins were extracted from mESC culture 24h in DM alone (gray bars) or in the presence of PTIO (striped bars). Data are shown as mean fold in-crase \pm s.e.m. compared to IgG value (striped bars) after Input normalization (n=3).

Fig 3c. Left panel: representative immunoprecipitation (IP) western blotting (WB) analysis of Hdac2 S-nitrosylation performed in mESC cultured in GS (24h) or after 2h from release into DM or NO. Right panel: the bar graph shows the mean result of relative densitometric analysis obtained from three independent experiments \pm s.e.m. (* $p < 0.05$ NO vs DM).

Fig 5d. Representative immunoprecipitation (IP) western blotting (WB) analysis of Hdac2 S-nitrosylation performed in mESC infected by Ad_GFP, Ad_eNOSwt or Ad_eNOS1177D cultured in GS or after 2h from release into DM (n=3 each group).

Fig 4a. qRT-PCR analysis of mRNA expression relative to the Wnt3, Wnt7b, Tbx4, Dll1 and Mesp2 transcripts in mES cultured for 24h in GS, DM and NO prior (control vector LCv2_NTC; black bar) and after CRISPR/Cas9 inactivation of Zeb1 (LCv2_Zeb1_1/_2; gray bars). Data are shown as the mean of three independent experiments for each CRISPR/Cas9

vector ± s.e.m. represented as fold increase compared to control mESC after subtraction of the housekeeping gene p0 signal (*p < 0.05 LCv2_Zeb1_1/_2 vs LCv2_NTC).

Suppl fig 5. Zeb1 overexpression contrasts mesendoermal marker expression in released mESC. (a) Representative western blotting analysis of cell extracts obtained from mESC after infection with lentivirus expressing GFP (Lenti_Empty) or Zeb1 (Lenti_Zeb1) cultured 24 h in GS, DM and NO probed with Zeb1 antibody. For each extract, α-tubulin was used as loading control. The experiment was repeated three times. (b) mRNA expression analysis of Wnt3, Wnt7b, Tbx4, Dll1 and Mesp2 in mESC after infection with lentivirus expressing GFP (Lenti_Empty) or Zeb1 (Lenti_Zeb1) cultured 24 h in GS, DM and NO. Data are represented as mean fold increase ± s.e.m. compared to GS_Lenti_Empty after subtraction of the housekeeping gene p0 signal (n=3; *p < 0.05 vs. GS_Lenti_Empty; °p < 0.05 vs. Lenti_Empty).

Fig 5b. qRT-PCR analysis of mRNA expression relative to the Wnt3, Dll1, Wnt7b, Mesp2, and Tbx4 transcripts in mESC cultured for 24 h in GS and DM prior (control vector LCv2_NTC; black bar) and after CRISPR/Cas9 inactivation of eNOS (LCv2_eNOS_1/_2; gray bars). Data are shown as the mean of three independent experiments for each CRISPR/Cas9 vector ± s.e.m. represented as fold increase compared to control mESC after subtraction of the housekeeping gene p0 signal (*p < 0.05 DM vs GS; °p < 0.05 LCv2_eNOS_1/_2 vs LCv2_NTC).

Q4. It may be possible to identify target genes that specifically lose Zeb1 or Hdac1/2 binding upon exposure to NO by performing ChIP-Seq for Zeb1 or Hdac1/2 in DM ± NO.

A4. In light of our data and considering the comments listed above the requested ChIP-seq analyses may not be technically feasible due to intrinsic limitations of the system. Specifically, in the differentiation medium with NO, one of the experimental condition requested by this Reviewer, the level of Zeb1 is barely detectable (see suppl. Fig. 2a-c below and in the manuscript body). Similarly, already in DM alone, Hdac2 appears nitrosylated and possibly inactivated by detachment from chromatin (see Fig. 3c and old Suppl. fig 3d (now Suppl. Fig. 3e) respectively; below and in the

manuscript body). Therefore it may be difficult to perform additional ChIP-seq analyses in the virtual absence of Zeb1 or Hdac2 from chromatin. Technically, in fact, in the presence of a reduced content of a chromatin-associated factor the efficiency of the chromatin-immunoprecipitation may decrease significantly preventing the sequencing reaction from performing efficiently with consequences on the number of reads and final data interpretation. For this reason we decided to utilize the more sensitive qPCR approach to reveal differences in Zeb1 or Hdac2 binding on the mesendodermal genes object of interest in this study.

Suppl fig 2. Zeb1 regulation in naïve and differentiating mESC. (a) Representative western blotting analysis of cell extracts obtained from mESC cultured 24h in SM, DM and NO (left panel) and GS, DM and NO (right panel) probed with Zeb1 antibody. For each extract, α -tubulin was used as loading control. The experiment was repeated three times. Right panels show the result of densitometric analysis ($*p < 0.05$ vs. stemness condition). **(b)** Representative confocal microscopy images depicting mESC cultured 24h in GS, DM or NO. Cells were probed by an anti-Zeb1 antibody (green; upper panels). Nuclei were counterstained with TROPRO 3 (blue; middle panels). Merged fluorescence images are shown in the lower panels. On the right, the insets contain enlargements of the selected areas (dashed squares). Magnification 40x. **(c)** Representative western blotting analysis of cytoplasm/nuclear fractionation extracts obtained from mESC cultured 24h in GS, DM or NO probed with Zeb1. For cytoplasmic extracts, α -tubulin was used as a loading control, whereas for nuclear extracts Fibrillarlin and H3 were used as loading control. The experiment was repeated three times.

Fig 3c. Left panel: representative immunoprecipitation (IP) western blotting (WB) analysis of Hdac2 S-nitrosylation performed in mESC cultured in GS (24h) or after 2h from release into DM or NO. Right panel: the bar graph shows the mean result of relative densitometric analysis obtained from three independent experiments \pm s.e.m. (* $p < 0.05$ NO vs DM).

Suppl fig 3d-now Suppl fig 3e. ChIP/qRT-PCR of Hdac2 chromatin association on the promoter regions of Wnt3, Wnt7b, Tbx4, Dll1 and Mesp2. Chromatins were extracted from mESC cultured 24 h in GS (black bars), DM (grey bars) or NO (white bars). Data are shown as mean fold increase \pm s.e.m. compared to IgG value after Input normalization. Three independent experiments were performed.

Q5. Following on the previous point, the authors should be careful throughout their study to distinguish between those effects mediated specifically by exposure to NO vs. effects that arise simply from the switch to DM. For example, Fig. 1c highlights differentially expressed genes (DEGs) between mESCs maintained in 2i/LIF (GS) vs. mESCs exposed to DM with exogenous NO. However, to determine the response to NO, it would be more appropriate to focus on DEGs in DM \pm NO. Similarly, the "NO up-genes" shown in Fig. 2a currently also include genes that are upregulated in DM alone, i.e. without exogenous NO.

A5. We thank the Reviewer for the interesting suggestion He/She raised in this revision. However, we must stress that our major interest here has been to investigate at molecular level the effect of NO in mESC, without distinguishing between endogenous production and exogenous administration. This line of research led to the identification of a NO positive cell sub-population recognized by NO production and mesendodermal gene expression already detectable in DM condition. The reason why some "NO up-genes" are upregulated already in DM is possibly ascribable to the endogenous production of NO as demonstrated by the L-NAME experiments shown in suppl. Fig. 4d (see below and in the manuscript body). In addition, the new data represented below indicate that the NO scavenger PTIO significantly reduces the expression of the signature genes in mESC released from pluripotency further corroborating the hypothesis that NO could play an important role in the mesendodermal differentiation program of these cells. However, following this reviewer suggestion new bioinformatics and gene ontology analyses were performed

on DEGs observed in NO vs DM. These results (see below) indicate that no clear pattern of differentiation lineage emerges from this comparison. Specifically apoptotic cell death pathways seem among the most upregulated ones while mixed neuroectodermal pathways are detectable among those underrepresented (see figure below). In light of these considerations, the authors feel that, although very important, the investigation of NO vs DM genes is more matter for future developments than object of specific attention at the present time.

Suppl fig 4d. mRNA expression analysis of eNOS, Wnt3, Dll1 and Wnt7b in mESC cultured 24 h in GS (black bars), DM (grey bars), DM + L-NAME (gray striped bars), DM + ODQ (gray squared bars) and NO (white bars). Data are represented as mean fold increase \pm s.e.m. compared to GS after subtraction of the housekeeping gene p0 signal (n=4; *p < 0.05 vs. GS; #p < 0.05 vs. DM).

New Suppl Fig 3b. mRNA expression analysis of Wnt3, Wnt7b, Tbx4, Dll1 and Mesp2 in mESC cultured 24 h in GS (black bars), DM (grey bars), DM + PTIO (gray striped bars). Data are represented as mean fold increase \pm s.e.m. compared to GS after subtraction of the housekeeping gene p0 signal (n=4).

Left panel: MA plot of differentially regulated genes (DEGs) expressed in mESC cultured for 24h in NO compared to DM condition. Red dots show genes with a padj < 0.05. Right panels: Gene ontology analysis of DEG between mESC cultured in NO and DM. Up-regulated genes depicted in the upper panel, red bar graph; down-regulated genes in the lower panel, blue bar graph.

Q6. In several panels the authors compare mRNA expression levels of five mesendodermal genes in mESCs in GS vs. 24h after exposure to DM or NO. However, the mRNA fold increases in Supplementary Figure 3a are much greater than those shown in Supplementary Figure 3e or Figure 4a-b. For example, Tbx4 was induced ~500-fold after 24h of NO treatment in Supplementary Figure 3a, but Tbx4 upregulation was only ~20-fold in Scramble-LNA control cells (Supplementary Figure 3e) or LCv2_NTC control cells (Figure 4a) at the same timepoint relative to GS. How do the authors explain these differences?

A6. We thank the Reviewer for pointing out this aspect. However we do not have a clear explanation for the diverse intensity of the responses observed for the same genes in the different experimental settings. We may only argue that in suppl. figure 3a cells were not significantly manipulated before assay while in suppl. Fig. 3e as well as in Fig. 4a-b cells underwent different treatments or rounds of selections. Specifically the results displayed in suppl. Fig. 3e were obtained after nucleofection while for those represented in figure 4a-b nucleofected and puromycin selected mESC were used. These specific manipulations may have affected the intensity of the response although no consequences were observed on the experimental trend of the biological effect. Statements specifying the different experimental conditions are present in the legend of suppl. figure 3e and/or in methods.

Q7. Some paragraphs are currently too long and contain too many details, which makes the paper inaccessible to a general audience. I recommend introducing line breaks at the following positions to split up the long paragraph in the Results section on p.9: lines 229-230 (start a new paragraph with “To investigate whether Zeb1 could be involved....”) and lines 253-254 (start a new paragraph with “The expression of the most represented NO-Zeb1-dependent genes...”).

A7. Manuscript can be edited according to Reviewer's suggestion.

Q8. The manuscript must be properly edited to improve English grammar and spelling. The Discussion alone contained the following obvious typos: "heterogeinicity" (line 396), "hypotesize" (line 435), "trascrition factor" (line 439), "repression transcription factor" (line 439, I suggest "repressive"), "metastatization" (line 464). The authors should also avoid terms like "stemness" (abstract, line 58) or "stemmed out" (Discussion, line 382), which are not very specific and instead refer to "self-renewal" or "pluripotency", where appropriate. The abbreviation "mES" is used throughout the text, but "ESC" or "ES cells" are more commonly used abbreviations.

A8. The manuscript language can be extensively revised according to suggestion.

Additional comments from reviewer #3:

I think the authors have offered a clear response to my questions. Their new data clarify that nitric oxide indeed triggers the eviction of Zeb1 and Hdac from chromatin, resulting in the upregulation of mesendodermal genes in embryonic stem cells. Therefore, I think they should be given an opportunity to resubmit a revised version of manuscript, provided that they include all of the new data and discussion points that are mentioned in the rebuttal.

Their revision should also address the following two points:

- 1) New Supplemental Figure 3a shows that mesendodermal transcripts are upregulated in differentiation medium, but much more significantly in the presence of exogenous NO. Therefore, the authors should acknowledge that exogenous NO seems to exert additional stimulation of mesendodermal target genes via mechanisms that are independent of Zeb1 or Hdac. At present the Discussion only mentions in passing that "NO donors are pleiotropic players" (p. 14) without directly referring to these data.

- 2) The new bioinformatic analysis comparing expression in NO vs. DM (Rebuttal, Answer #5) suggests that exogenous NO downregulates genes associated with neural development. This is an interesting finding and suggests that NO may also promote mesendodermal differentiation by blocking transition to alternative cell fates.

Additional comments from reviewer #3:

Q1. I think the authors have offered a clear response to my questions. Their new data clarify that nitric oxide indeed triggers the eviction of Zeb1 and Hdac from chromatin, resulting in the upregulation of mesendodermal genes in embryonic stem cells. Therefore, I think they should be given an opportunity to resubmit a revised version of manuscript, provided that they include all of the new data and discussion points that are mentioned in the rebuttal.

A1. The authors would like to thank the Reviewer for His/Her positive evaluation to their point-to-point letter. New data are provided as Supplementary Fig. 2; new version of Supplementary Fig. 4a; and Supplementary Fig.4b, respectively. Related results have been described in the Result section and commented in the Discussion of the revised manuscript (pages 7-9 lines 178-189 and 230-288; pages 12-13 lines 351-363). All discussion points mentioned in the rebuttal letter have been included in the present version of the manuscript (pages 12-13)

Q2. New Supplemental Figure 3a shows that mesendodermal transcripts are upregulated in differentiation medium, but much more significantly in the presence of exogenous NO. Therefore, the authors should acknowledge that exogenous NO seems to exert additional stimulation of mesendodermal target genes via mechanisms that are independent of Zeb1 or Hdac. At present the Discussion only mentions in passing that "NO donors are pleiotropic players" (p. 14) without directly referring to these data.

A2. We thank the reviewer for His/Her valuable suggestion. A paragraph has been added in the Discussion section showing/recognizing the contribution of exogenous NO to exert increased induction of mesendodermal target genes in the revised manuscript (pages 12-13 lines 343-362).

Q3. The new bioinformatic analysis comparing expression in NO vs. DM (Rebuttal, Answer #5) suggests that exogenous NO downregulates genes associated with neural development. This is an interesting finding and suggests that NO may also promote mesendodermal differentiation by blocking transition to alternative cell fates.

A3. The new bioinformatics analysis derived from NO/DM comparison has been added to the revised manuscript as Supplementary Fig.2 and exploited to further confirm that NO favours mesendodermal differentiation by preventing the transition to alternative cell fates like neuroectodermal one. This evidence has been commented at pages 7-8 line 187-189 of the present version of the manuscript.